# GO GRADIENT FOR EXPECTATION-BASED OBJECTIVES

**Yulai Cong**[*]      **Miaoyun Zhao**[*]      **Ke Bai**      **Lawrence Carin**
Department of Electrical and Computer Engineering, Duke University

## ABSTRACT

Within many machine learning algorithms, a fundamental problem concerns efficient calculation of an unbiased gradient wrt parameters $\boldsymbol{\gamma}$ for expectation-based objectives $\mathbb{E}_{q_{\boldsymbol{\gamma}}(\boldsymbol{y})}[f(\boldsymbol{y})]$. Most existing methods either ($i$) suffer from high variance, seeking help from (often) complicated variance-reduction techniques; or ($ii$) they only apply to reparameterizable continuous random variables and employ a reparameterization trick. To address these limitations, we propose a General and One-sample (GO) gradient that ($i$) applies to many distributions associated with non-reparameterizable continuous *or* discrete random variables, and ($ii$) has the same low-variance as the reparameterization trick. We find that the GO gradient often works well in practice based on only one Monte Carlo sample (although one can of course use more samples if desired). Alongside the GO gradient, we develop a means of propagating the chain rule through distributions, yielding statistical back-propagation, coupling neural networks to common random variables.

## 1 INTRODUCTION

Neural networks, typically trained using back-propagation for parameter optimization, have recently demonstrated significant success across a wide range of applications. There has been interest in coupling neural networks with random variables, so as to embrace greater descriptive capacity. Recent examples of this include black-box variational inference (BBVI) (Kingma & Welling, 2014; Rezende et al., 2014; Ranganath et al., 2014; Hernández-Lobato et al., 2016; Ranganath et al., 2016b; Li & Turner, 2016; Ranganath et al., 2016a; Zhang et al., 2018) and generative adversarial networks (GANs) (Goodfellow et al., 2014; Radford et al., 2015; Zhao et al., 2016; Arjovsky et al., 2017; Li et al., 2017; Gan et al., 2017; Li et al., 2018). Unfortunately, efficiently backpropagating gradients through general distributions (random variables) remains a bottleneck. Most current methodology focuses on distributions with continuous random variables, for which the reparameterization trick may be readily applied (Kingma & Welling, 2014; Grathwohl et al., 2017).

As an example, the aforementioned bottleneck greatly constrains the applicability of BBVI, by limiting variational approximations to reparameterizable distributions. This limitation excludes discrete random variables and many types of continuous ones. From the perspective of GAN, the need to employ reparameterization has constrained most applications to continuous observations. There are many forms of data that are more-naturally discrete.

The fundamental problem associated with the aforementioned challenges is the need to efficiently calculate an unbiased low-variance gradient wrt parameters $\boldsymbol{\gamma}$ for an expectation objective of the form $\mathbb{E}_{q_{\boldsymbol{\gamma}}(\boldsymbol{y})}[f(\boldsymbol{y})]$[1]. We are interested in general distributions $q_{\boldsymbol{\gamma}}(\boldsymbol{y})$, for which the components of $\boldsymbol{y}$ may be either continuous or discrete. Typically the components of $\boldsymbol{y}$ have a hierarchical structure, and a subset of the components of $\boldsymbol{y}$ play a role in evaluating $f(\boldsymbol{y})$.

Unfortunately, classical methods for estimating gradients of $\mathbb{E}_{q_{\boldsymbol{\gamma}}(\boldsymbol{y})}[f(\boldsymbol{y})]$ wrt $\boldsymbol{\gamma}$ have limitations. The REINFORCE gradient (Williams, 1992), although generally applicable (*e.g.*, for continuous and discrete random variables), exhibits high variance with Monte Carlo (MC) estimation of the

---

[*]Correspondence to: Yulai Cong <yulaicong@gmail.com>, Miaoyun Zhao <miaoyun9zhao@gmail.com>.

[1]In this paper, we consider expectation objectives meeting basic assumptions: ($i$) $q_{\boldsymbol{\gamma}}(\boldsymbol{y})$ is differentiable wrt $\boldsymbol{\gamma}$; ($ii$) $f(\boldsymbol{y})$ is differentiable for continuous $\boldsymbol{y}$; and ($iii$) $f(\boldsymbol{y}) < \infty$ for discrete $\boldsymbol{y}$. For simplicity of the main paper, those assumptions are implicitly made, as well as fundamental rules like Leibniz's rule.

expectation, forcing one to apply additional variance-reduction techniques. The reparameterization trick (Rep) (Salimans et al., 2013; Kingma & Welling, 2014; Rezende et al., 2014) works well, with as few as only one MC sample, but it is limited to continuous reparameterizable $\boldsymbol{y}$. Many efforts have been devoted to improving these two formulations, as detailed in Section 6. However, none of these methods is characterized by generalization (applicable to general distributions) and efficiency (working well with as few as one MC sample).

The key contributions of this work are based on the recognition that REINFORCE and Rep are seeking to solve the same objective, but in practice Rep yields lower-variance estimations, albeit for a narrower class of distributions. Recent work (Ranganath et al., 2016b) has made a connection between REINFORCE and Rep, recognizing that the former estimates a term the latter evaluates analytically. The high variance by which REINFORCE approximates this term manifests high variance in the gradient estimation. Extending these ideas, we make the following main contributions. ($i$) We propose a new General and One-sample (GO) gradient in Section 3, that principally generalizes Rep to many non-reparameterizable distributions and justifies two recent methods (Figurnov et al., 2018; Jankowiak & Obermeyer, 2018); the "One sample" motivating the name GO is meant to highlight the low variance of the proposed method, although of course one may use more than one sample if desired. ($ii$) We find that the core of the GO gradient is something we term a *variable-nabla*, which can be interpreted as the gradient of a random variable wrt a parameter. ($iii$) Utilizing variable-nablas to propagate the chain rule through distributions, we broaden the applicability of the GO gradient in Sections 4-5 and present statistical back-propagation, a statistical generalization of classic back-propagation (Rumelhart & Hinton, 1986). Through this generalization, we may couple neural networks to general random variables, and compute needed gradients with low variance.

## 2 BACKGROUND

To motivate this paper, we begin by briefly elucidating common machine learning problems for which there is a need to efficiently estimate gradients of $\boldsymbol{\gamma}$ for functions of the form $\mathbb{E}_{q_{\boldsymbol{\gamma}}(\boldsymbol{y})}[f(\boldsymbol{y})]$. Assume access to data samples $\{\boldsymbol{x}_i\}_{i=1,N}$, drawn i.i.d. from the true (and unknown) underlying distribution $q(\boldsymbol{x})$. We seek to learn a model $p_{\boldsymbol{\theta}}(\boldsymbol{x})$ to approximate $q(\boldsymbol{x})$. A classic approach to such learning is to maximize the expected log likelihood $\hat{\boldsymbol{\theta}} = \text{argmax}_{\boldsymbol{\theta}} \, \mathbb{E}_{q(\boldsymbol{x})}[\log p_{\boldsymbol{\theta}}(\boldsymbol{x})]$, perhaps with an added regularization term on $\boldsymbol{\theta}$. Expectation $\mathbb{E}_{q(\boldsymbol{x})}(\cdot)$ is approximated via the available data samples, as $\hat{\boldsymbol{\theta}} = \text{argmax}_{\boldsymbol{\theta}} \frac{1}{N} \sum_{i=1}^{N} \log p_{\boldsymbol{\theta}}(\boldsymbol{x}_i)$.

It is often convenient to employ a model with latent variables $\boldsymbol{z}$, i.e., $p_{\boldsymbol{\theta}}(\boldsymbol{x}) = \int p_{\boldsymbol{\theta}}(\boldsymbol{x}, \boldsymbol{z}) d\boldsymbol{z} = \int p_{\boldsymbol{\theta}}(\boldsymbol{x}|\boldsymbol{z})p(\boldsymbol{z})d\boldsymbol{z}$, with prior $p(\boldsymbol{z})$ on $\boldsymbol{z}$. The integral wrt $\boldsymbol{z}$ is typically intractable, motivating introduction of the approximate posterior $q_{\boldsymbol{\phi}}(\boldsymbol{z}|\boldsymbol{x})$, with parameters $\boldsymbol{\phi}$. The well-known evidence lower bound (ELBO) (Jordan et al., 1999; Bishop, 2006) is defined as

$$
\begin{aligned}
\text{ELBO}(\boldsymbol{\theta}, \boldsymbol{\phi}; \boldsymbol{x}) &= \mathbb{E}_{q_{\boldsymbol{\phi}}(\boldsymbol{z}|\boldsymbol{x})}[\log p_{\boldsymbol{\theta}}(\boldsymbol{x}, \boldsymbol{z}) - \log q_{\boldsymbol{\phi}}(\boldsymbol{z}|\boldsymbol{x})] & (1) \\
&= \log p_{\boldsymbol{\theta}}(\boldsymbol{x}) - \text{KL}[q_{\boldsymbol{\phi}}(\boldsymbol{z}|\boldsymbol{x})\|p_{\boldsymbol{\theta}}(\boldsymbol{z}|\boldsymbol{x})] \leq \log p_{\boldsymbol{\theta}}(\boldsymbol{x}) & (2)
\end{aligned}
$$

where $p_{\boldsymbol{\theta}}(\boldsymbol{z}|\boldsymbol{x})$ is the true posterior, and $\text{KL}(\cdot\|\cdot)$ represents the Kullback-Leibler divergence. Variational learning seeks $(\hat{\boldsymbol{\theta}}, \hat{\boldsymbol{\phi}}) = \text{argmax}_{\boldsymbol{\theta}, \boldsymbol{\phi}} \sum_{i=1}^{N} \text{ELBO}(\boldsymbol{\theta}, \boldsymbol{\phi}; \boldsymbol{x}_i)$.

While computation of the ELBO has been considered for many years, a problem introduced recently concerns *adversarial* learning of $p_{\boldsymbol{\theta}}(\boldsymbol{x})$, or, more precisely, learning a model that allows one to efficiently and accurately draw samples $\boldsymbol{x} \sim p_{\boldsymbol{\theta}}(\boldsymbol{x})$ that are similar to $\boldsymbol{x} \sim q(\boldsymbol{x})$. With generative adversarial networks (GANs) (Goodfellow et al., 2014), one seeks to solve

$$
\min_{\boldsymbol{\theta}} \max_{\boldsymbol{\beta}} \left\{ \mathbb{E}_{q(\boldsymbol{x})}[\log D_{\boldsymbol{\beta}}(\boldsymbol{x})] + \mathbb{E}_{p_{\boldsymbol{\theta}}(\boldsymbol{x})}[\log(1 - D_{\boldsymbol{\beta}}(\boldsymbol{x}))] \right\}, \tag{3}
$$

where $D_{\boldsymbol{\beta}}(\boldsymbol{x})$ is a discriminator with parameters $\boldsymbol{\beta}$, quantifying the probability $\boldsymbol{x}$ was drawn from $q(\boldsymbol{x})$, with $1 - D_{\boldsymbol{\beta}}(\boldsymbol{x})$ representing the probability that it was drawn from $p_{\boldsymbol{\theta}}(\boldsymbol{x})$. There have been many recent extensions of GAN (Radford et al., 2015; Zhao et al., 2016; Arjovsky et al., 2017; Li et al., 2017; Gan et al., 2017; Li et al., 2018), but the basic setup in (3) holds for most.

To optimize (1) and (3), the most challenging gradients that must be computed are of the form $\nabla_{\boldsymbol{\gamma}} \mathbb{E}_{q_{\boldsymbol{\gamma}}(\boldsymbol{y})}[f(\boldsymbol{y})]$; for (1) $\boldsymbol{y} = \boldsymbol{z}$ and $\boldsymbol{\gamma} = \boldsymbol{\phi}$, while for (3) $\boldsymbol{y} = \boldsymbol{x}$ and $\boldsymbol{\gamma} = \boldsymbol{\theta}$. The need to evaluate expressions like $\nabla_{\boldsymbol{\gamma}} \mathbb{E}_{q_{\boldsymbol{\gamma}}(\boldsymbol{y})}[f(\boldsymbol{y})]$ arises in many other machine learning problems, and consequently it has generated much prior attention.

Evaluation of the gradient of the expectation is simplified if the gradient can be moved inside the expectation. REINFORCE (Williams, 1992) is based on the identity

$$\nabla_{\boldsymbol{\gamma}}\mathbb{E}_{q_{\boldsymbol{\gamma}}(\boldsymbol{y})}[f(\boldsymbol{y})] = \mathbb{E}_{q_{\boldsymbol{\gamma}}(\boldsymbol{y})}\big[f(\boldsymbol{y})\nabla_{\boldsymbol{\gamma}}\log q_{\boldsymbol{\gamma}}(\boldsymbol{y})\big]. \tag{4}$$

While simple in concept, this estimate is known to have high variance when the expectation $\mathbb{E}_{q_{\boldsymbol{\gamma}}(\boldsymbol{y})}(\cdot)$ is approximated (as needed in practice) by a finite number of samples.

An approach (Salimans et al., 2013; Kingma & Welling, 2014; Rezende et al., 2014) that has attracted recent attention is termed the reparameterization trick, applicable when $q_{\boldsymbol{\gamma}}(\boldsymbol{y})$ can be reparametrized as $\boldsymbol{y} = \boldsymbol{\tau}_{\boldsymbol{\gamma}}(\boldsymbol{\epsilon})$, with $\boldsymbol{\epsilon} \sim q(\boldsymbol{\epsilon})$, where $\boldsymbol{\tau}_{\boldsymbol{\gamma}}(\boldsymbol{\epsilon})$ is a differentiable transformation and $q(\boldsymbol{\epsilon})$ is a simple distribution that may be readily sampled. To keep consistency with the literature, we call a distribution reparameterizable if and only if those conditions are satisfied[2]. In this case we have

$$\nabla_{\boldsymbol{\gamma}}\mathbb{E}_{q_{\boldsymbol{\gamma}}(\boldsymbol{y})}[f(\boldsymbol{y})] = \mathbb{E}_{q(\boldsymbol{\epsilon})}\big[[\nabla_{\boldsymbol{\gamma}}\boldsymbol{\tau}_{\boldsymbol{\gamma}}(\boldsymbol{\epsilon})][\nabla_{\boldsymbol{y}}f(\boldsymbol{y})]|_{\boldsymbol{y}=\boldsymbol{\tau}_{\boldsymbol{\gamma}}(\boldsymbol{\epsilon})}\big]. \tag{5}$$

This gradient, termed Rep, is typically characterized by relatively low variance, when approximating $\mathbb{E}_{q(\boldsymbol{\epsilon})}(\cdot)$ with a small number of samples $\boldsymbol{\epsilon} \sim q(\boldsymbol{\epsilon})$. This approach has been widely employed for computation of the ELBO and within GAN, but it limits one to models that satisfy the assumptions of Rep.

## 3 GO GRADIENT

The reparameterization trick (Rep) is limited to reparameterizable random variables $\boldsymbol{y}$ with continuous components. There are situations for which Rep is not readily applicable, *e.g.*, where the components of $\boldsymbol{y}$ may be discrete or nonnegative Gamma distributed. We seek to gain insights from the relationship between REINFORCE and Rep, and generalize the types of random variables $\boldsymbol{y}$ for which the latter approach may be effected. We term our proposed approach a General and One-sample (GO) gradient. In practice, we find that this approach works well with as few as one sample for evaluating the expectation, and it is applicable to more general settings than Rep.

Recall that Rep was first applied within the context of variational learning (Kingma & Welling, 2014), as in (1). Specifically, it was assumed $q_{\boldsymbol{\gamma}}(\boldsymbol{y}) = \prod_v q_{\boldsymbol{\gamma}}(y_v)$, omitting explicit dependence on data $\boldsymbol{x}$, for notational convenience; $y_v$ is component $v$ of $\boldsymbol{y}$. In Kingma & Welling (2014) $q_{\boldsymbol{\gamma}}(y_v)$ corresponded to a Gaussian distribution $q_{\boldsymbol{\gamma}}(y_v) = \mathcal{N}(y_v; \mu_v(\boldsymbol{\gamma}), \sigma_v^2(\boldsymbol{\gamma}))$, with mean $\mu_v(\boldsymbol{\gamma})$ and variance $\sigma_v^2(\boldsymbol{\gamma})$. In the following we generalize $q_{\boldsymbol{\gamma}}(y_v)$ such that it need not be Gaussian. Applying integration by parts (Ranganath et al., 2016b)

$$\nabla_{\boldsymbol{\gamma}}\mathbb{E}_{q_{\boldsymbol{\gamma}}(\boldsymbol{y})}[f(\boldsymbol{y})] = \sum_v \mathbb{E}_{q_{\boldsymbol{\gamma}}(\boldsymbol{y}_{-v})}\Big[\int f(\boldsymbol{y})\nabla_{\boldsymbol{\gamma}}q_{\boldsymbol{\gamma}}(y_v)dy_v\Big] \tag{6}$$

$$= \sum_v \mathbb{E}_{q_{\boldsymbol{\gamma}}(\boldsymbol{y}_{-v})}\Big[\underbrace{[f(\boldsymbol{y})\nabla_{\boldsymbol{\gamma}}Q_{\boldsymbol{\gamma}}(y_v)]\big|_{-\infty}^{\infty}}_{\text{``0''}}\underbrace{-\int[\nabla_{\boldsymbol{\gamma}}Q_{\boldsymbol{\gamma}}(y_v)][\nabla_{y_v}f(\boldsymbol{y})]dy_v}_{\text{``Key''}}\Big], \tag{7}$$

where $\boldsymbol{y}_{-v}$ denotes $\boldsymbol{y}$ with $y_v$ excluded, and $Q_{\boldsymbol{\gamma}}(y_v)$ is the cumulative distribution function (CDF) of $q_{\boldsymbol{\gamma}}(y_v)$. The "0" term is readily proven to be zero for any $Q_{\boldsymbol{\gamma}}(y_v)$, with the assumption that $f(\boldsymbol{y})$ doesn't tend to infinity faster than $\nabla_{\boldsymbol{\gamma}}Q_{\boldsymbol{\gamma}}(y_v)$ tending to zero when $y_v \to \pm\infty$.

The "Key" term exactly recovers the *one-dimensional* Rep when reparameterization $y_v = \tau_{\boldsymbol{\gamma}}(\epsilon_v)$, $\epsilon_v \sim q(\epsilon_v)$ exists (Ranganath et al., 2016b). Further, applying $\nabla_{\boldsymbol{\gamma}}q_{\boldsymbol{\gamma}}(y_v) = q_{\boldsymbol{\gamma}}(y_v)\nabla_{\boldsymbol{\gamma}}\log q_{\boldsymbol{\gamma}}(y_v)$ in (6) yields REINFORCE. Consequently, it appears that Rep yields low variance by analytically setting to zero the unnecessary but high-variance-injecting "0" term, while in contrast REINFORCE implicitly seeks to numerically implement both terms in (7).

We generalize $q_{\boldsymbol{\gamma}}(\boldsymbol{y})$ for discrete $y_v$, here assuming $y_v \in \{0, 1, \ldots, \infty\}$. It is shown in Appendix A.2 that this framework is also applicable to discrete $y_v$ with a *finite* alphabet. It may be shown (see Appendix A.2) that

$$\nabla_{\boldsymbol{\gamma}}\mathbb{E}_{q_{\boldsymbol{\gamma}}(\boldsymbol{y})}[f(\boldsymbol{y})] = \sum_v \mathbb{E}_{q_{\boldsymbol{\gamma}}(\boldsymbol{y}_{-v})}\Big[\sum_{y_v} f(\boldsymbol{y})\nabla_{\boldsymbol{\gamma}}q_{\boldsymbol{\gamma}}(y_v)\Big]$$

$$= \sum_v \mathbb{E}_{q_{\boldsymbol{\gamma}}(\boldsymbol{y}_{-v})}\Big[\underbrace{[f(\boldsymbol{y})\nabla_{\boldsymbol{\gamma}}Q_{\boldsymbol{\gamma}}(y_v)]|_{y_v=\infty}}_{\text{``0''}}\underbrace{-\sum_{y_v}[\nabla_{\boldsymbol{\gamma}}Q_{\boldsymbol{\gamma}}(y_v)][f(\boldsymbol{y}_{-v}, y_v+1) - f(\boldsymbol{y})]}_{\text{``Key''}}\Big] \tag{8}$$

---

[2] A Bernoulli random variable $y \sim \text{Bernoulli}(P)$ is identical to $y = 1_{\epsilon<P}$ with $\epsilon \sim U(0, 1)$. But $y$ is called non-reparameterizable because the transformation $y = 1_{\epsilon<P}$ is not differentiable.

where $Q_{\boldsymbol{\gamma}}(y_v) = \sum_{n=0}^{y_v} q_{\boldsymbol{\gamma}}(n)$, and $Q_{\boldsymbol{\gamma}}(\infty) = 1$ for all $\boldsymbol{\gamma}$.

**Theorem 1** (**GO Gradient**). *For expectation objectives* $\mathbb{E}_{q_{\boldsymbol{\gamma}}(\boldsymbol{y})}[f(\boldsymbol{y})]$, *where* $q_{\boldsymbol{\gamma}}(\boldsymbol{y})$ *satisfies (i)* $q_{\boldsymbol{\gamma}}(\boldsymbol{y}) = \prod_v q_{\boldsymbol{\gamma}}(y_v)$; *(ii) the corresponding CDF* $Q_{\boldsymbol{\gamma}}(y_v)$ *is differentiable wrt parameters* $\boldsymbol{\gamma}$; *and (iii) one can calculate* $\nabla_{\boldsymbol{\gamma}} Q_{\boldsymbol{\gamma}}(y_v)$, *the General and One-sample (GO) gradient is defined as*

$$\nabla_{\boldsymbol{\gamma}} \mathbb{E}_{q_{\boldsymbol{\gamma}}(\boldsymbol{y})}[f(\boldsymbol{y})] = \mathbb{E}_{q_{\boldsymbol{\gamma}}(\boldsymbol{y})}\Big[\mathbb{G}_{\boldsymbol{\gamma}}^{q_{\boldsymbol{\gamma}}(\boldsymbol{y})} \mathbb{D}_{\boldsymbol{y}}[f(\boldsymbol{y})]\Big], \tag{9}$$

*where* $\mathbb{G}_{\boldsymbol{\gamma}}^{q_{\boldsymbol{\gamma}}(\boldsymbol{y})}$ *specifies* $\mathbb{G}_{\boldsymbol{\kappa}}^{q(\boldsymbol{y})} \triangleq \big[\cdots, g_{\boldsymbol{\kappa}}^{q(y_v)}, \cdots\big]$ *with variable-nabla* $g_{\boldsymbol{\kappa}}^{q(y_v)} \triangleq \frac{-1}{q(y_v)} \nabla_{\boldsymbol{\kappa}} Q(y_v)$, $\mathbb{D}_{\boldsymbol{y}}[f(\boldsymbol{y})] = \big[\cdots, \mathbb{D}_{y_v}[f(\boldsymbol{y})], \cdots\big]^T$, *and*

$$\mathbb{D}_{y_v}[f(\boldsymbol{y})] \triangleq \begin{cases} \nabla_{y_v} f(\boldsymbol{y}), & \textit{Continous } y_v \\ f(\boldsymbol{y}_{-v}, y_v + 1) - f(\boldsymbol{y}), & \textit{Discrete } y_v \end{cases}$$

All proofs are provided in Appendix A, where we also list $g_{\boldsymbol{\gamma}}^{q_{\boldsymbol{\gamma}}(y_v)}$ for a wide selection of possible $q_{\boldsymbol{\gamma}}(y)$, for both continuous and discrete $y$. Note for the special case with continuous $\boldsymbol{y}$, GO reduces to Implicit Rep gradients (Figurnov et al., 2018) and pathwise derivatives (Jankowiak & Obermeyer, 2018); in other words, GO provides a principled explanation for their low variance, namely their foundation (implicit differentiation) originates from integration by parts. For high-dimensional discrete $\boldsymbol{y}$, calculating $\mathbb{D}_{\boldsymbol{y}}[f(\boldsymbol{y})]$ is computationally expensive. Fortunately, for $f(\boldsymbol{y})$ often used in practice special properties hold that can be exploited for efficient parallel computing. Also for discrete $y_v$ with finite support, it is possible that one could analytically evaluate a part of expectations in (9) for lower variance, mimicking the local idea in Titsias & Lázaro-Gredilla (2015); Titsias (2015). Appendix I shows an example illustrating how to handle these two issues in practice.

## 4 DEEP GO GRADIENT

The GO gradient in Theorem 1 can only handle single-layer mean-field $q_{\boldsymbol{\gamma}}(\boldsymbol{y})$, characterized by an independence assumption on the components of $\boldsymbol{y}$. One may enlarge the descriptive capability of $q_{\boldsymbol{\gamma}}(\boldsymbol{y})$ by modeling it as a marginal distribution of a deep model (Ranganath et al., 2016b; Bishop, 2006). Hereafter, we focus on this situation, and begin with a 2-layer model for simple demonstration. Specifically, consider

$$q_{\boldsymbol{\gamma}}(\boldsymbol{y}) = \int q_{\boldsymbol{\gamma}}(\boldsymbol{y}, \boldsymbol{\lambda}) d\boldsymbol{\lambda} = \int q_{\boldsymbol{\gamma}_{\boldsymbol{y}}}(\boldsymbol{y}|\boldsymbol{\lambda}) q_{\boldsymbol{\gamma}_{\boldsymbol{\lambda}}}(\boldsymbol{\lambda}) d\boldsymbol{\lambda} = \int \prod_v q_{\boldsymbol{\gamma}_{\boldsymbol{y}}}(y_v|\boldsymbol{\lambda}) \cdot \prod_k q_{\boldsymbol{\gamma}_{\boldsymbol{\lambda}}}(\lambda_k) d\boldsymbol{\lambda},$$

where $\boldsymbol{\gamma} = \{\boldsymbol{\gamma}_{\boldsymbol{y}}, \boldsymbol{\gamma}_{\boldsymbol{\lambda}}\}$, $\boldsymbol{y}$ is the *leaf* variable, and the *internal* variable $\boldsymbol{\lambda}$ is assumed to be continuous. Components of $\boldsymbol{y}$ are assumed to be *conditionally* independent given $\boldsymbol{\lambda}$, but upon marginalizing out $\boldsymbol{\lambda}$ this independence is removed.

The objective becomes $\mathbb{E}_{q_{\boldsymbol{\gamma}}(\boldsymbol{y})}[f(\boldsymbol{y})] = \mathbb{E}_{q_{\boldsymbol{\gamma}_{\boldsymbol{y}}}(\boldsymbol{y}|\boldsymbol{\lambda}) q_{\boldsymbol{\gamma}_{\boldsymbol{\lambda}}}(\boldsymbol{\lambda})}[f(\boldsymbol{y})]$, and via Theorem 1

$$\nabla_{\boldsymbol{\gamma}_{\boldsymbol{y}}} \mathbb{E}_{q_{\boldsymbol{\gamma}}(\boldsymbol{y})}[f(\boldsymbol{y})] = \mathbb{E}_{q_{\boldsymbol{\gamma}}(\boldsymbol{y}, \boldsymbol{\lambda})}\Big[\mathbb{G}_{\boldsymbol{\gamma}_{\boldsymbol{y}}}^{q_{\boldsymbol{\gamma}_{\boldsymbol{y}}}(\boldsymbol{y}|\boldsymbol{\lambda})} \mathbb{D}_{\boldsymbol{y}}[f(\boldsymbol{y})]\Big]. \tag{10}$$

**Lemma 1.** *Equation* (10) *exactly recovers the Rep gradient in* (5), *if* $\boldsymbol{\gamma} = \boldsymbol{\gamma}_{\boldsymbol{y}}$ *and* $q_{\boldsymbol{\gamma}}(\boldsymbol{y})$ *has reparameterization* $\boldsymbol{y} = \boldsymbol{\tau}_{\boldsymbol{\gamma}}(\boldsymbol{\lambda})$, $\boldsymbol{\lambda} \sim q(\boldsymbol{\lambda})$ *for differentiable* $\boldsymbol{\tau}_{\boldsymbol{\gamma}}(\boldsymbol{\lambda})$ *and easily sampled* $q(\boldsymbol{\lambda})$.

Lemma 1 shows that Rep is a special case of our deep GO gradient in the following Theorem 2. Note neither Implicit Rep gradients nor pathwise derivatives can recover Rep in general, because a neural-network-parameterized $\boldsymbol{y} = \boldsymbol{\tau}_{\boldsymbol{\gamma}}(\boldsymbol{\lambda})$ may lead to non-trivial CDF $Q_{\boldsymbol{\gamma}}(\boldsymbol{y})$.

For the gradient wrt $\boldsymbol{\gamma}_{\boldsymbol{\lambda}}$, we first apply Theorem 1, yielding

$$\nabla_{\boldsymbol{\gamma}_{\boldsymbol{\lambda}}} \mathbb{E}_{q_{\boldsymbol{\gamma}}(\boldsymbol{y})}[f(\boldsymbol{y})] = \mathbb{E}_{q_{\boldsymbol{\gamma}_{\boldsymbol{\lambda}}}(\boldsymbol{\lambda})}\Big[\mathbb{G}_{\boldsymbol{\gamma}_{\boldsymbol{\lambda}}}^{q_{\boldsymbol{\gamma}_{\boldsymbol{\lambda}}}(\boldsymbol{\lambda})} \mathbb{D}_{\boldsymbol{\lambda}}\big[\mathbb{E}_{q_{\boldsymbol{\gamma}_{\boldsymbol{y}}}(\boldsymbol{y}|\boldsymbol{\lambda})}[f(\boldsymbol{y})]\big]\Big].$$

For continuous internal variable $\boldsymbol{\lambda}$ one can apply Theorem 1 again, from which

$$\nabla_{\boldsymbol{\gamma}_{\boldsymbol{\lambda}}} \mathbb{E}_{q_{\boldsymbol{\gamma}}(\boldsymbol{y})}[f(\boldsymbol{y})] = \mathbb{E}_{q_{\boldsymbol{\gamma}}(\boldsymbol{y}, \boldsymbol{\lambda})}\Big[\mathbb{G}_{\boldsymbol{\gamma}_{\boldsymbol{\lambda}}}^{q_{\boldsymbol{\gamma}_{\boldsymbol{\lambda}}}(\boldsymbol{\lambda})} \mathbb{G}_{\boldsymbol{\lambda}}^{q_{\boldsymbol{\gamma}_{\boldsymbol{y}}}(\boldsymbol{y}|\boldsymbol{\lambda})} \mathbb{D}_{\boldsymbol{y}}[f(\boldsymbol{y})]\Big]. \tag{11}$$

Now extending the same procedure to deeper models with $L$ layers, we generalize the GO gradient in Theorem 2. Random variable $\boldsymbol{y}^{(L)}$ is assumed to be the leaf variable of interest, and may be continuous or discrete; latent/internal random variables $\{\boldsymbol{y}^{(1)}, \ldots, \boldsymbol{y}^{(L-1)}\}$ are assumed continuous (these generalize $\boldsymbol{\lambda}$ from above).

**Theorem 2 (Deep GO Gradient).** *For expectation objectives $\mathbb{E}_{q_{\boldsymbol{\gamma}}(\boldsymbol{y}^{(L)})}[f(\boldsymbol{y}^{(L)})]$ with $q_{\boldsymbol{\gamma}}(\boldsymbol{y}^{(L)})$ being the marginal distribution of*

$$q_{\boldsymbol{\gamma}}\big(\boldsymbol{y}^{(1)}, \cdots, \boldsymbol{y}^{(L)}\big) = q_{\boldsymbol{\gamma}^{(1)}}\big(\boldsymbol{y}^{(1)}\big)\Big[\prod\nolimits_{l=2}^{L} q_{\boldsymbol{\gamma}^{(l)}}\big(\boldsymbol{y}^{(l)}|\boldsymbol{y}^{(l-1)}\big)\Big],$$

*where $\boldsymbol{\gamma} = \{\boldsymbol{\gamma}^{(1)}, \cdots, \boldsymbol{\gamma}^{(L)}\}$, all internal variables $\boldsymbol{y}^{(1)}, \cdots, \boldsymbol{y}^{(L-1)}$ are continuous, the leaf variable $\boldsymbol{y}^{(L)}$ is either continuous or discrete, $q_{\boldsymbol{\gamma}^{(l)}}(\boldsymbol{y}^{(l)}|\boldsymbol{y}^{(l-1)}) = \prod_v q_{\boldsymbol{\gamma}^{(l)}}(y_v^{(l)}|\boldsymbol{y}^{(l-1)})$, and one has access to variable-nablas $g_{\boldsymbol{\gamma}^{(l)}}^{q_{\boldsymbol{\gamma}^{(l)}}(y_v^{(l)}|\boldsymbol{y}^{(l-1)})}$ and $g_{\boldsymbol{y}^{(l-1)}}^{q_{\boldsymbol{\gamma}^{(l)}}(y_v^{(l)}|\boldsymbol{y}^{(l-1)})}$, as defined in Theorem 1, the General and One-sample (GO) gradient is defined as*

$$\nabla_{\boldsymbol{\gamma}^{(l)}} \mathbb{E}_{q_{\boldsymbol{\gamma}}(\boldsymbol{y}^{(L)})}[f(\boldsymbol{y}^{(L)})] = \mathbb{E}_{q_{\boldsymbol{\gamma}}(\boldsymbol{y}^{(1)}, \cdots, \boldsymbol{y}^{(L)})}\Big[\mathbb{G}_{\boldsymbol{\gamma}^{(l)}}^{q_{\boldsymbol{\gamma}^{(l)}}(\boldsymbol{y}^{(l)}|\boldsymbol{y}^{(l-1)})} \mathbb{BP}[\boldsymbol{y}^{(l)}]\Big], \qquad (12)$$

*where $\mathbb{BP}[\boldsymbol{y}^{(L)}] = \mathbb{D}_{\boldsymbol{y}^{(L)}}[f(\boldsymbol{y}^{(L)})]$ and $\mathbb{BP}[\boldsymbol{y}^{(l)}] = \mathbb{G}_{\boldsymbol{y}^{(l)}}^{q_{\boldsymbol{\gamma}^{(l+1)}}(\boldsymbol{y}^{(l+1)}|\boldsymbol{y}^{(l)})} \mathbb{BP}[\boldsymbol{y}^{(l+1)}]$ for $l < L$.*

**Corollary 1.** *The deep GO gradient in Theorem 2 exactly recovers back-propagation (Rumelhart & Hinton, 1986) when each element distribution $q_{\boldsymbol{\gamma}^{(l)}}(y_v^{(l)}|\boldsymbol{y}^{(l-1)})$ is specified as the Dirac delta function located at the activated value after activation function.*

Figure 1 is presented for better understanding. With variable-nablas, one can readily verify that the deep GO gradient in Theorem 2 *in expectation* obeys the chain rule.

Figure 1: Relating back-propagation (Rumelhart & Hinton, 1986) with the deep GO gradient in Theorem 2. (*i*) In deterministic deep neural networks, one forward-propagates information using activation functions, like ReLU, to sequentially activate $\{\boldsymbol{y}^{(l)}\}_{l=1,\cdots,L}$ (black solid arrows), and then back-propagates gradients from Loss $f(\cdot)$ to each parameter $\boldsymbol{\gamma}^{(l)}$ via gradient-flow through $\{\boldsymbol{y}^{(k)}\}_{k=L,\cdots,l}$ (red dashed arrows). (*ii*) Similarly for the deep GO gradient with 1 MC sample, one forward-propagates information to calculate the expected loss function $f(\boldsymbol{y}^{(L)})$ using distributions as statistical activation functions, and then uses variable-nablas to sequentially back-propagate gradients through random variables $\{\boldsymbol{y}^{(k)}\}_{k=L,\cdots,l}$ to each $\boldsymbol{\gamma}^{(l)}$, as in (12).

# 5 STATISTICAL BACK-PROPAGATION AND HIERARCHICAL VARIATIONAL INFERENCE

Recall the motivating discussion in Section 2, in which we considered generative model $p_{\boldsymbol{\theta}}(\boldsymbol{x}, \boldsymbol{z})$ and inference model $q_{\boldsymbol{\phi}}(\boldsymbol{z}|\boldsymbol{x})$, the former used to model synthesis of the observed data $\boldsymbol{x}$ and the latter used for inference of $\boldsymbol{z}$ given observed $\boldsymbol{x}$. In recent deep architectures, a hierarchical representation for cumulative latent variables $\boldsymbol{z} = (\boldsymbol{z}^{(1)}, \ldots, \boldsymbol{z}^{(L)})$ has been considered (Rezende et al., 2014; Ranganath et al., 2015; Zhou et al., 2015; 2016; Ranganath et al., 2016b; Cong et al., 2017; Zhang et al., 2018). As an example, there are models with $p_{\boldsymbol{\theta}}(\boldsymbol{x}, \boldsymbol{z}) = p_{\boldsymbol{\theta}}(\boldsymbol{x}|\boldsymbol{z}^{(1)})\big[\prod_{l=1}^{L-1} p_{\boldsymbol{\theta}}(\boldsymbol{z}^{(l)}|\boldsymbol{z}^{(l+1)})\big]p_{\boldsymbol{\theta}}(\boldsymbol{z}^{(L)})$. When performing inference for such models, it is intuitive to consider first-order Markov chain structure for $q_{\boldsymbol{\phi}}(\boldsymbol{z}|\boldsymbol{x}) = q_{\boldsymbol{\phi}}(\boldsymbol{z}^{(1)}|\boldsymbol{x})\prod_{l=2}^{L} q_{\boldsymbol{\phi}}(\boldsymbol{z}^{(l)}|\boldsymbol{z}^{(l-1)})$. The discussion in this section is most relevant for variational inference, for computation of $\nabla_{\boldsymbol{\phi}} \mathbb{E}_{q_{\boldsymbol{\phi}}(\boldsymbol{z}^{(1)},\ldots,\boldsymbol{z}^{(L)}|\boldsymbol{x})}[f(\boldsymbol{z}^{(1)}, \ldots, \boldsymbol{z}^{(L)})]$, and consequently we specialize to that notation in the subsequent discussion (we consider representations in terms of $\boldsymbol{z}$, rather than the more general $\boldsymbol{y}$ notation employed in Section 4).

Before proceeding, we seek to make clear the distinction between this section and Section 4. In the latter, only the leaf variable $\boldsymbol{z}^{(L)}$ appears in $\nabla_{\boldsymbol{\gamma}} \mathbb{E}_{q_{\boldsymbol{\gamma}}(\boldsymbol{z}^{(L)})}[f(\boldsymbol{z}^{(L)})]$; see (12), with $\boldsymbol{y}^{(L)} \to \boldsymbol{z}^{(L)}$. That

is because in Section 4 the underlying model is a marginal distribution of $\boldsymbol{z}^{(L)}$, $i.e.$, $q_{\boldsymbol{\gamma}}(\boldsymbol{z}^{(L)})$, which is relevant to the generators of GANs; see (3), with $\boldsymbol{x} \rightarrow \boldsymbol{z}^{(L)}$ and $p_{\boldsymbol{\theta}}(\boldsymbol{x}) \rightarrow q_{\boldsymbol{\gamma}}(\boldsymbol{z}^{(L)})$. Random variables $\boldsymbol{z}^{(1)}, \ldots, \boldsymbol{z}^{(L-1)}$ were marginalized out of $q_{\boldsymbol{\gamma}}(\boldsymbol{z}^{(1)}, \ldots, \boldsymbol{z}^{(L-1)}, \boldsymbol{z}^{(L)})$ to represent $q_{\boldsymbol{\gamma}}(\boldsymbol{z}^{(L)})$. As discussed in Section 4, $\boldsymbol{z}^{(1)}, \ldots, \boldsymbol{z}^{(L-1)}$ were added there to enhance the modeling flexibility of $q_{\boldsymbol{\gamma}}(\boldsymbol{z}^{(L)})$. In this section, the deep set of random variables $\boldsymbol{z} = (\boldsymbol{z}^{(1)}, \ldots, \boldsymbol{z}^{(L)})$ are inherent components of the underlying generative model for $\boldsymbol{x}$, $i.e.$, $p_{\boldsymbol{\theta}}(\boldsymbol{x}, \boldsymbol{z}) = p_{\boldsymbol{\theta}}(\boldsymbol{x}, \boldsymbol{z}^{(1)}, \ldots, \boldsymbol{z}^{(L)})$. Hence, all components of $\boldsymbol{z}$ manifested via inference model $q_{\boldsymbol{\phi}}(\boldsymbol{z}|\boldsymbol{x}) = q_{\boldsymbol{\phi}}(\boldsymbol{z}^{(1)}, \ldots, \boldsymbol{z}^{(L)}|\boldsymbol{x})$ play a role in $f(\boldsymbol{z})$. Besides, no specific structure is imposed on $p_{\boldsymbol{\theta}}(\boldsymbol{x}, \boldsymbol{z})$ and $q_{\boldsymbol{\phi}}(\boldsymbol{z}|\boldsymbol{x})$ in this section, moving beyond the aforementioned first-order Markov structure. For a practical application, one may employ domain knowledge to design suitable graphical models for $p_{\boldsymbol{\theta}}(\boldsymbol{x}, \boldsymbol{z})$ and $q_{\boldsymbol{\phi}}(\boldsymbol{z}|\boldsymbol{x})$, and then use the following Theorem 3 for training.

**Theorem 3 (Statistical Back-Propagation).** *For expectation objectives*

$$\mathbb{E}_{q_{\boldsymbol{\phi}}(\{\boldsymbol{z}^{(i)}\}_{i=1}^L)}\big[f(\{\boldsymbol{z}^{(i)}\}_{i=1}^L)\big],$$

*where $\{\boldsymbol{z}^{(i)}\}_{i=1}^I$ denotes $I$ continuous internal variables with at least one child variable, $\{\boldsymbol{z}^{(j)}\}_{j=I+1}^L$ represents $L - I$ continuous or discrete leaf variables with no children except $f(\cdot)$, and $q_{\boldsymbol{\phi}}(\cdot)$ is constructed as a hierarchical probabilistic graphical model*

$$q_{\boldsymbol{\phi}}(\{\boldsymbol{z}^{(i)}\}_{i=1}^L) = q_{\boldsymbol{\phi}}(\{\boldsymbol{z}^{(i)}\}_{i=1}^I) \prod_{j=I+1}^L q_{\boldsymbol{\phi}}(\boldsymbol{z}^{(j)}|\{\boldsymbol{z}^{(i)}\}_{i=1}^I)$$

*with each element distribution $q_{\boldsymbol{\phi}}(z_v|pa(z_v))$ having accessible variable-nablas as defined in Theorem 1, $pa(z_v)$ denotes the parent variables of $z_v$, the General and One-sample (GO) gradient for $\boldsymbol{\phi}_k \in \boldsymbol{\phi}$ is defined as*

$$\nabla_{\boldsymbol{\phi}_k} \mathbb{E}_{q_{\boldsymbol{\phi}}(\{\boldsymbol{z}^{(i)}\}_{i=1}^L)}\big[f(\{\boldsymbol{z}^{(i)}\}_{i=1}^L)\big] = \mathbb{E}_{q_{\boldsymbol{\phi}}(\{\boldsymbol{z}^{(i)}\}_{i=1}^L)}\Big[\mathbb{G}_{\boldsymbol{\phi}_k}^{q_{\boldsymbol{\phi}}[ch(\boldsymbol{\phi}_k)]}\mathbb{BP}[ch(\boldsymbol{\phi}_k)]\Big], \qquad (13)$$

*where $ch(\boldsymbol{\phi}_k)$ denotes the children variables of $\boldsymbol{\phi}_k$, and with $z_v \in ch(\boldsymbol{\phi}_k)$,*

$$\mathbb{G}_{\boldsymbol{\phi}_k}^{q_{\boldsymbol{\phi}}[ch(\boldsymbol{\phi}_k)]} = [\cdots, g_{\boldsymbol{\phi}_k}^{q_{\boldsymbol{\phi}}(z_v|pa(z_v))}, \cdots], \qquad \mathbb{BP}[ch(\boldsymbol{\phi}_k)] = [\cdots, \mathbb{BP}[z_v], \cdots]^T,$$

*and $\mathbb{BP}[z_v]$ is iteratively calculated as $\mathbb{BP}[z_v] = \mathbb{G}_{z_v}^{q_{\boldsymbol{\phi}}[ch(z_v)]}\mathbb{BP}[ch(z_v)]$, until leaf variables where $\mathbb{BP}[z_v] = \mathbb{D}_{z_v}[f(\{\boldsymbol{z}^{(i)}\}_{i=1}^L)]$.*

Statistical back-propagation in Theorem 3 is relevant to hierarchical variational inference (HVI) (Ranganath et al., 2016b; Hoffman & Blei, 2015; Mnih & Gregor, 2014) (see Appendix G), greatly generalizing GO gradients to the inference of directed acyclic probabilistic graphical models. In HVI variational distributions are specified as hierarchical graphical models constructed by neural networks. Using statistical back-propagation, one may rely on GO gradients to perform HVI with low variance, while greatly broadening modeling flexibility.

## 6 RELATED WORK

There are many methods directed toward low-variance gradients for expectation-based objectives. Attracted by the generalization of REINFORCE, many works try to improve its performance via efficient variance-reduction techniques, like control variants (Mnih & Gregor, 2014; Titsias & Lázaro-Gredilla, 2015; Gu et al., 2015; Mnih & Rezende, 2016; Tucker et al., 2017; Grathwohl et al., 2017) or via data augmentation and permutation techniques (Yin & Zhou, 2018). Most of this research focuses on discrete random variables, likely because Rep (if it exists) works well for continuous random variables but it may not exist for discrete random variables. Other efforts are devoted to continuously relaxing discrete variables, to combine both REINFORCE and Rep for variance reduction (Jang et al., 2016; Maddison et al., 2016; Tucker et al., 2017; Grathwohl et al., 2017).

Inspired by the low variance of Rep, there are methods that try to generalize its scope. The Generalized Rep (GRep) gradient (Ruiz et al., 2016) employs an approximate reparameterization whose transformed distribution weakly depends on the parameters of interest. Rejection sampling variational inference (RSVI) (Naesseth et al., 2016) exploits highly-tuned transformations in mature rejection sampling simulation to better approximate Rep for non-reparameterizable distributions. Compared

to the aforementioned methods, the proposed GO gradient, containing Rep as a special case for continuous random variables, applies to both continuous and discrete random variables with the same low-variance as the Rep gradient. Implicit Rep gradients (Figurnov et al., 2018) and pathwise derivatives (Jankowiak & Obermeyer, 2018) are recent low-variance methods that exploit the gradient of the expected function; they are special cases of GO in the single-layer continuous settings.

The idea of gradient backpropagation through random variables has been exploited before. RE-LAX (Grathwohl et al., 2017), employing neural-network-parametrized control variants to assist REINFORCE for that goal, has a variance *potentially* as low as the Rep gradient. SCG (Schulman et al., 2015) utilizes the generalizability of REINFORCE to construct widely-applicable stochastic computation graphs. However, REINFORCE is known to have high variance, especially for high-dimensional problems, where the proposed methods are preferable when applicable (Schulman et al., 2015). Stochastic back-propagation (Rezende et al., 2014; Fan et al., 2015), focusing mainly on reparameterizable Gaussian random variables and deep latent Gaussian models, exploits the product rule for an integral to derive gradient backpropagation through several continuous random variables. By comparison, the proposed statistical back-propagation based on the GO gradient is applicable to most distributions for continuous random variables. Further, it also flexibly generalizes to hierarchical probabilistic graphical models with continuous internal variables and continuous/discrete leaf ones.

## 7 EXPERIMENTS

We examine the proposed GO gradients and statistical back-propagation with four experiments: ($i$) simple one-dimensional (gamma and negative binomial) examples are presented to verify the GO gradient in Theorem 1, corresponding to nonnegative and discrete random variables; ($ii$) the discrete variational autoencoder experiment from Tucker et al. (2017) and Grathwohl et al. (2017) is reproduced to compare GO with the state-of-the-art variance-reduction methods; ($iii$) a multinomial GAN, generating discrete observations, is constructed to demonstrate the deep GO gradient in Theorem 2; ($iv$) hierarchical variational inference (HVI) for two deep non-conjugate Bayesian models is developed to verify statistical back-propagation in Theorem 3. Note the experiments of Figurnov et al. (2018) and Jankowiak & Obermeyer (2018) additionally support our GO in the single-layer continuous settings.

Many mature machine learning frameworks, like TensorFlow (Abadi et al.) and PyTorch (Paszke et al., 2017), are optimized for implementation of methods like back-propagation. Fortunately, all gradient calculations in the proposed theorems obey the chain rule in expectation, enabling convenient incorporation of the proposed approaches into existing frameworks. Experiments presented below were implemented in TensorFlow or PyTorch with a Titan Xp GPU. Code for all experiments can be found at `github.com/YulaiCong/GOgradient`.

**Notation** $\mathrm{Gam}(\alpha, \beta)$ denotes the gamma distribution with shape $\alpha$ and rate $\beta$, $\mathrm{NB}(r, P)$ the negative binomial distribution with number of failures $r$ and success probability $P$, $\mathrm{Bern}(P)$ the Bernoulli distribution with probability $P$, $\mathrm{Mult}(n, \boldsymbol{P})$ the multinomial distribution with number of trials $n$ and event probabilities $\boldsymbol{P}$, $\mathrm{Pois}(\lambda)$ the Poisson distribution with rate $\lambda$, and $\mathrm{Dir}(\boldsymbol{\alpha})$ the Dirichlet distribution with concentration parameters $\boldsymbol{\alpha}$.

### 7.1 GAMMA AND NB ONE-DIMENSIONAL SIMPLE EXAMPLES

We first consider illustrative *one-dimensional* "toy" problems, to examine the GO gradient for both continuous and discrete random variables. The optimization objective is expressed as

$$\max_{\boldsymbol{\phi}} \mathrm{ELBO}(\boldsymbol{\phi}) = \mathbb{E}_{q_{\boldsymbol{\phi}}(z)}[\log p(z|x) - \log q_{\boldsymbol{\phi}}(z)] + \log p(x),$$

where for continuous $z$ we assume $p(z|x) = \mathrm{Gam}(z; \alpha_0, \beta_0)$ for given set $(\alpha_0, \beta_0)$, with $q_{\boldsymbol{\phi}}(z) = \mathrm{Gam}(z; \alpha, \beta)$ and $\boldsymbol{\phi} = \{\alpha, \beta\}$; for discrete $z$ we assume $p(z|x) = \mathrm{NB}(z; r_0, p_0)$ for given set $(r_0, p_0)$, with $q_{\boldsymbol{\phi}}(z) = \mathrm{NB}(z; r, p)$ and $\boldsymbol{\phi} = \{r, p\}$. Stochastic gradient ascent with one-sample-estimated gradients is used to optimize the objective, which is equivalent to minimizing $\mathrm{KL}(q_{\boldsymbol{\phi}}(z) \| p(z|x))$.

Figure 2 shows the results (see Appendix H for additional details). For the nonnegative continuous $z$ associated with the gamma distribution, we compare our GO gradient with GRep (Ruiz et al., 2016),

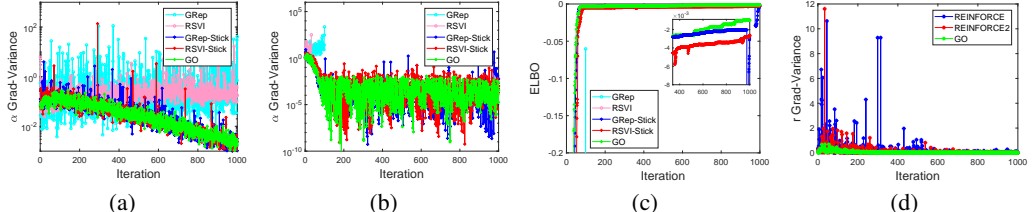

Figure 2: Gamma (a-c) and NB (d) toy experimental results. (a) The gradient variance of gamma shape $\alpha$ versus iterations, with posterior parameters $\alpha_0 = 1, \beta_0 = 0.5$. (b)-(c) The gradient variance of $\alpha$ and ELBOs versus iterations respectively, when $\alpha_0 = 0.01, \beta_0 = 0.5$. (d) The gradient variance of NB $r$ versus iterations with $r_0 = 10, p_0 = 0.2$. In each iteration, gradient variances are estimated with 20 Monte Carlo samples (each sample corresponds to one gradient estimate), among which the last one is used to update parameters.

RSVI (Naesseth et al., 2016), and their modified version using the "sticking" idea (Roeder et al., 2017), denoted as GRep-Stick and RSVI-Stick, respectively. For RSVI and RSVI-Stick, the shape augmentation parameter is set as 5 by default. The only difference between GRep and GRep-Stick (also RSVI and RSVI-Stick) is the latter does *not* analytically express the entropy $\mathbb{E}_{q_\phi(z)}[-\log q_\phi(z)]$. Figure 2(a) clearly shows the utility of employing sticking to reduce variance; without it, GRep and RSVI exhibit high variance, that destabilizes the optimization for small gamma shape parameters, as shown in Figures 2(b) and 2(c). We adopt the sticking approach hereafter for all the compared methods. Among methods with sticking, GO exhibits the lowest variance in general, as shown in Figures 2(a) and 2(b). GO empirically provides more stable learning curves, as shown in Figure 2(c). For the discrete case corresponding to the NB distribution, GO is compared to REINFORCE (Williams, 1992). To address the concern about comparison with the same number of evaluations of the expected function[3], another curve of REINFORCE using 2 samples is also added, termed REINFORCE2. It is apparent from Figure 2(d) that, thanks to analytically removing the "0" terms in (8), the GO gradient has much lower variance, even in this simple one-dimensional case.

## 7.2 DISCRETE VARIATIONAL AUTOENCODER

To demonstrate the low variance of the proposed GO gradient, we consider the discrete variational autoencoder (VAE) experiment from REBAR (Tucker et al., 2017) and RELAX (Grathwohl et al., 2017), to make a direct comparison with state-of-the-art variance-reduction methods. Since the statistical back-propagation in Theorem 3 cannot handle discrete internal variables, we focus on the single-latent-layer settings (1 layer of 200 Bernoulli random variables), *i.e.,*

$$p_\theta(\boldsymbol{x}, \boldsymbol{z}) : \boldsymbol{x} \sim \text{Bern}\big(\text{NN}_{\boldsymbol{P}_{\boldsymbol{x}|\boldsymbol{z}}}(\boldsymbol{z})\big), \boldsymbol{z} \sim \text{Bern}\big(\boldsymbol{P}_{\boldsymbol{z}}\big)$$
$$q_\phi(\boldsymbol{z}|\boldsymbol{x}) : \boldsymbol{z} \sim \text{Bern}\big(\text{NN}_{\boldsymbol{P}_{\boldsymbol{z}|\boldsymbol{x}}}(\boldsymbol{x})\big)$$

where $\boldsymbol{P}_{\boldsymbol{z}}$ is the parameters of the prior $p_\theta(\boldsymbol{z})$, $\text{NN}_{\boldsymbol{P}_{\boldsymbol{x}|\boldsymbol{z}}}(\boldsymbol{z})$ means using a neural network to project the latent binary code $\boldsymbol{z}$ to the parameters $\boldsymbol{P}_{\boldsymbol{x}|\boldsymbol{z}}$ of the likelihood $p_\theta(\boldsymbol{x}|\boldsymbol{z})$, and $\text{NN}_{\boldsymbol{P}_{\boldsymbol{z}|\boldsymbol{x}}}(\boldsymbol{x})$ is similarly defined for $q_\phi(\boldsymbol{z}|\boldsymbol{x})$. The objective is given in (1). See Appendix I for more details.

Table 1: Best obtained ELBOs for discrete variational autoencoders. Results of REBAR and RELAX are obtained by running the released code[4] from Grathwohl et al. (2017). All methods are run with the same learning rate for $1,000,000$ iterations.

| Dataset | Model | Training | | | Validation | | |
|---|---|---|---|---|---|---|---|
| | | REBAR | RELAX | GO | REBAR | RELAX | GO |
| **MNIST** | Linear 1 layer | -112.16 | -112.89 | **-110.21** | -114.85 | -115.36 | **-114.27** |
| | Nonlinear | -96.99 | -95.99 | **-82.26** | -112.96 | -112.42 | **-111.48** |
| **Omniglot** | Linear 1 layer | -122.19 | -122.17 | **-119.03** | -124.81 | -124.95 | **-123.84** |
| | Nonlinear | -79.51 | -80.67 | **-54.96** | -129.00 | -128.99 | **-126.59** |

---

[3]In the NB experiments, REINFORCE uses 1 sample and 1 evaluation of the expected function; REINFORCE2 uses 2 sample and 2 evaluations; and GO uses 1 sample and 2 evaluations.

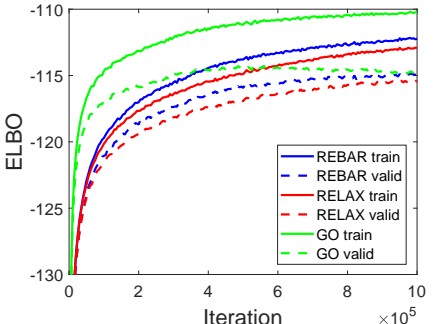 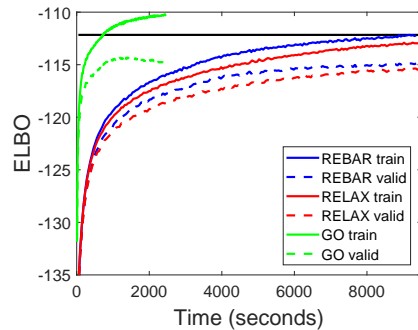

Figure 3: Training curves for the discrete VAE experiment with 1-layer linear model (see Appendix I) on the stochastically binarized MNIST dataset(Salakhutdinov & Murray, 2008). All methods are run with the same learning rate for $1,000,000$ iterations. The black line represents the best training ELBO of REBAR and RELAX. ELBOs are calculated using all training/validation data.

Table 2: Inception scores on quantized MNIST. BGAN's results are run with the author-released code https://github.com/rdevon/BGAN.

| bits (states) | BGAN | MNGAN-GO |
|---|---|---|
| 1 (1) | $8.31 \pm .06$ | $\mathbf{9.10 \pm .06}$ |
| 1 (2) | $\mathbf{8.56 \pm .04}$ | $8.40 \pm .07$ |
| 2 (4) | $7.76 \pm .04$ | $\mathbf{9.02 \pm .06}$ |
| 3 (8) | $7.26 \pm .03$ | $\mathbf{9.26 \pm .07}$ |
| 4 (16) | $6.29 \pm .05$ | $\mathbf{9.27 \pm .06}$ |

Figure 4: Generated samples from BGAN (top) and MNGAN-GO (bottom) trained on 4-bit quantized MNIST.

Figure 3 (also Figure 9 of Appendix I) shows the training curves versus iteration and running time for the compared methods. Even without any variance-reduction techniques, GO provides better performance, faster convergence rate, and better running efficiency (about ten times faster in achieving the best training ELBO of RERAR/RELAX in this experiment). We believe GO's better performance originates from: ($i$) its inherent low-variance nature; ($ii$) GO has less parameters compared to REBAR and RELAX (no control variant is adopted for GO); ($iii$) efficient batch processing methods (see Appendix I) are adopted to benefit from parallel computing. Table 1 presents the best training/validation ELBOs under various experimental settings for the compared methods. GO provides the best performance in all situations. Additional experimental results are given in Appendix I.

Many variance-reduction techniques can be used to further reduce the variance of GO, especially when complicated models are of interest. Compared to RELAX, GO cannot be directly applied when $f(\boldsymbol{y})$ is not computable or where the interested model has discrete internal variables (like multilayer Sigmoid belief networks (Neal, 1992)). For the latter issue, we present in Appendix B.4 a procedure to assist GO (or statistical back-propagation in Theorem 3) in handling discrete internal variables.

### 7.3 MULTINOMIAL GAN

To demonstrate the deep GO gradient in Theorem 2, we adopt multinomial leaf variables $\boldsymbol{x}$ and construct a new multinomial GAN (denoted as MNGAN-GO) for generating discrete observations with a finite alphabet. The corresponding generator $p_{\boldsymbol{\theta}}(\boldsymbol{x})$ is expressed as

$$\boldsymbol{\epsilon} \sim \mathcal{N}(\mathbf{0}, \mathbf{I}), \boldsymbol{x} \sim \mathrm{Mult}(\mathbf{1}, \mathrm{NN}_{\mathbf{P}}(\boldsymbol{\epsilon})).$$

For brevity, we integrate the generator's parameters $\boldsymbol{\theta}$ into the NN notation, and do not explicitly express them. Details for this example are provided in Appendix J.

We compare MNGAN-GO with the recently proposed boundary-seeking GAN (BGAN) (Hjelm et al., 2018) on 1-bit (1-state, Bernoulli leaf variables $\boldsymbol{x}$), 1-bit (2-state), 2-bit (4-state), 3-bit (8-state) and

---

[4]`github.com/duvenaud/relax`

4-bit (16-state) discrete image generation tasks, using quantized MNIST datasets (LeCun et al., 1998). Table 2 presents inception scores (Salimans et al., 2016) of both methods. MNGAN-GO performs better in general. Further, with GO's assistance, MNGAN-GO shows more potential to benefit from richer information coming from more quantized states. For demonstration, Figure 4 shows the generated samples from the 4-bit experiment, where better image quality and higher diversity are observed for the samples from MNGAN-GO.

## 7.4 HVI FOR DEEP EXPONENTIAL FAMILIES AND DEEP LATENT DIRICHLET ALLOCATION

To demonstrate statistical back-propagation in Theorem 3, we design variational inference nets for two nonconjugate hierarchical Bayesian models, *i.e.,* deep exponential families (DEF) (Ranganath et al., 2015) and deep latent Dirichlet allocation (DLDA) (Zhou et al., 2015; 2016; Cong et al., 2017).

$$\text{DEF} : \boldsymbol{x} \sim \text{Pois}(\mathbf{W}^{(1)}\boldsymbol{z}^{(1)}), \boldsymbol{z}^{(l)} \sim \text{Gam}\big(\alpha_z, {}^{\alpha_z}/\mathbf{w}^{(l+1)}\boldsymbol{z}^{(l+1)}\big), \mathbf{W}^{(l)} \sim \text{Gam}(\alpha_0, \beta_0)$$

$$\text{DLDA} : \boldsymbol{x} \sim \text{Pois}(\boldsymbol{\Phi}^{(1)}\boldsymbol{z}^{(1)}), \boldsymbol{z}^{(l)} \sim \text{Gam}\big(\boldsymbol{\Phi}^{(l+1)}\boldsymbol{z}^{(l+1)}, c^{(l+1)}\big), \boldsymbol{\Phi}^{(l)} \sim \text{Dir}(\eta_0).$$

For demonstration, we design the inference nets $q_{\boldsymbol{\phi}}(\boldsymbol{z}|\boldsymbol{x})$ following the first-order Markov chain construction in Section 5, namely

$$q_{\boldsymbol{\phi}}(\boldsymbol{z}|\boldsymbol{x}) : \boldsymbol{z}^{(1)} \sim \text{Gam}\big(\text{NN}_{\boldsymbol{\alpha}}^{(1)}(\boldsymbol{x}), \text{NN}_{\boldsymbol{\beta}}^{(1)}(\boldsymbol{x})\big), \boldsymbol{z}^{(l)} \sim \text{Gam}\big(\text{NN}_{\boldsymbol{\alpha}}^{(l)}(\boldsymbol{z}^{(l-1)}), \text{NN}_{\boldsymbol{\beta}}^{(l)}(\boldsymbol{z}^{(l-1)})\big).$$

Further details are provided in Appendix K. One might also wish to design inference nets that have structure beyond the above first-order Markov chain construction, as in Zhang et al. (2018); we do not consider that here, but Theorem 3 is applicable to that case.

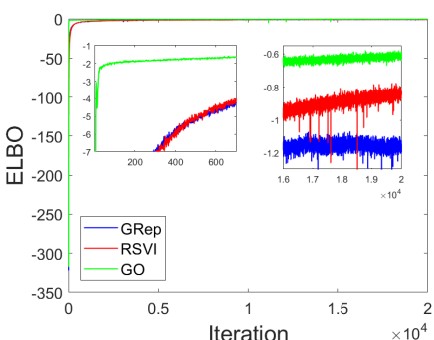

Figure 5: ELBOs of HVI for a 128-64 DEF on MNIST. Except for different ways to calculate gradients, all other experimental settings are the same for compared methods, including the sticking idea and one-MC-sample gradient estimate.

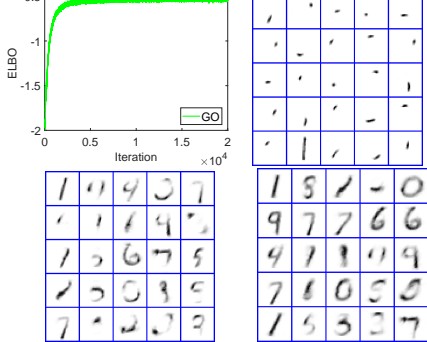

Figure 6: HVI results for a 128-64-32 DLDA on MNIST. Upperleft is the training ELBOs. The remaining subfigures are learned dictionary atoms from $\boldsymbol{\Phi}^{(1)}$ (top-right), $\boldsymbol{\Phi}^{(1)}\boldsymbol{\Phi}^{(2)}$ (bottom-left), and $\boldsymbol{\Phi}^{(1)}\boldsymbol{\Phi}^{(2)}\boldsymbol{\Phi}^{(3)}$ (bottom-right).

HVI for a 2-layer DEF is first performed, with the ELBO curves shown in Figure 5. GO enables faster and more stable convergence. Figure 6 presents the HVI results for a 3-layer DLDA, for which stable ELBOs are again observed. More importantly, with the GO gradient, one can utilize pure gradient-based methods to efficiently train such complicated nonconjugate models for meaningful dictionaries (see Appendix K for more implementary details).

## 8 CONCLUSIONS

For expectation-based objectives, we propose a General and One-sample (GO) gradient that applies to continuous and discrete random variables. We further generalize the GO gradient to cases for which the underlying model is deep and has a marginal distribution corresponding to the latent variables of interest, and to cases for which the latent variables are hierarchical. The GO-gradient setup is demonstrated to yield the same low-variance estimation as the reparameterization trick, which is only applicable to reparameterizable continuous random variables. Alongside the GO gradient, we constitute a means of propagating the chain rule through distributions. Accordingly, we present statistical back-propagation, to flexibly integrate deep neural networks with general classes of random variables.

ACKNOWLEDGMENTS

We thank the anonymous reviewers for their useful comments. The research was supported by part by DARPA, DOE, NIH, NSF and ONR. The Titan Xp GPU used was donated by the NVIDIA Corporation. We also wish to thank Chenyang Tao, Liqun Chen, and Chunyuan Li for helpful discussions.

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

## A    PROOF OF THEOREM 1

We first prove (7) in the main manuscript, followed by its discrete counterpart, *i.e.,* (8) in the main manuscript. Then, it is easy to verify Theorem 1.

### A.1    PROOF OF EQUATION (7) IN THE MAIN MANUSCRIPT

Similar proof in one-dimension is also given in the supplemental materials of Ranganath et al. (2016b).

We want to calculate

$$\nabla_{\boldsymbol{\gamma}}\mathbb{E}_{q_{\boldsymbol{\gamma}}(\boldsymbol{y})}[f(\boldsymbol{y})] = \nabla_{\boldsymbol{\gamma}}\mathbb{E}_{\prod_v q_{\boldsymbol{\gamma}}(y_v)}[f(\boldsymbol{y})] = \sum_v \mathbb{E}_{q_{\boldsymbol{\gamma}}(\boldsymbol{y}_{-v})}\big[\textstyle\int f(\boldsymbol{y})\nabla_{\boldsymbol{\gamma}}q_{\boldsymbol{\gamma}}(y_v)dy_v\big],$$

where $\boldsymbol{y}_{-v}$ denotes $\boldsymbol{y}$ with $y_v$ excluded. Without loss of generality, we assume $y_v \in (-\infty, \infty)$.

Let $v'(y_v) = \nabla_{\boldsymbol{\gamma}}q_{\boldsymbol{\gamma}}(y_v)$, and we have

$$v(y_v) = \int_{-\infty}^{y_v}\nabla_{\boldsymbol{\gamma}}q_{\boldsymbol{\gamma}}(t)dt = \nabla_{\boldsymbol{\gamma}}\int_{-\infty}^{y_v}q_{\boldsymbol{\gamma}}(t)dt = \nabla_{\boldsymbol{\gamma}}Q_{\boldsymbol{\gamma}}(y_v),$$

where $Q_{\boldsymbol{\gamma}}(y_v)$ is the cumulative distribution function (CDF) of $q_{\boldsymbol{\gamma}}(y_v)$. Further define $u(y_v) = f(y_v, \boldsymbol{y}_{-v})$, we then apply integration by parts (or partial integration) to get

$$
\begin{aligned}
\nabla_{\boldsymbol{\gamma}}\mathbb{E}_{q_{\boldsymbol{\gamma}}(\boldsymbol{y})}[f(\boldsymbol{y})] &= \sum_v \mathbb{E}_{q_{\boldsymbol{\gamma}}(\boldsymbol{y}_{-v})}\big[\textstyle\int u(y_v)v'(y_v)dy_v\big] \\
&= \sum_v \mathbb{E}_{q_{\boldsymbol{\gamma}}(\boldsymbol{y}_{-v})}\big[u(y_v)v(y_v)|_{-\infty}^{\infty} - \textstyle\int u'(y_v)v(y_v)dy_v\big] \\
&= \sum_v \mathbb{E}_{q_{\boldsymbol{\gamma}}(\boldsymbol{y}_{-v})}\big[\underbrace{f(\boldsymbol{y})\nabla_{\boldsymbol{\gamma}}Q_{\boldsymbol{\gamma}}(y_v)|_{-\infty}^{\infty}}_{\text{``0''}} \underbrace{- \textstyle\int[\nabla_{\boldsymbol{\gamma}}Q_{\boldsymbol{\gamma}}(y_v)][\nabla_{y_v}f(\boldsymbol{y})]dy_v}_{\text{``Key''}}\big].
\end{aligned}
\tag{14}
$$

With $Q_{\boldsymbol{\gamma}}(\infty) = 1$ and $Q_{\boldsymbol{\gamma}}(-\infty) = 0$, it's straightforward to verify that the first term is always zero for any $Q_{\boldsymbol{\gamma}}(y_v)$, thus named the "0" term.

### A.2    PROOF OF EQUATION (8) IN THE MAIN MANUSCRIPT

For discrete variables $\boldsymbol{y}$, we have

$$\nabla_{\boldsymbol{\gamma}}\mathbb{E}_{q_{\boldsymbol{\gamma}}(\boldsymbol{y})}[f(\boldsymbol{y})] = \sum_v \mathbb{E}_{q_{\boldsymbol{\gamma}}(\boldsymbol{y}_{-v})}\Big[\sum_{y_v=0}^{N} f(\boldsymbol{y})\nabla_{\boldsymbol{\gamma}}q_{\boldsymbol{\gamma}}(y_v)\Big],$$

where $y_v \in \{0, 1, \cdots, N\}$ and $N$ is the size of the alphabet.

To handle the summation of products of two sequences and develop discrete counterpart of (7), we first introduce Abel transformation.
**Abel transformation.** Given two sequences $\{a_n\}$ and $\{b_n\}$, with $n \in \{0, \cdots, N\}$, we define $B_0 = b_0$ and $B_n = \sum_{k=0}^{n} b_n$ for $n \geq 1$. Accordingly, we have

$$
\begin{aligned}
S_N = \sum_{n=0}^{N} a_n b_n &= a_0 b_0 + \sum_{n=1}^{N} a_n(B_n - B_{n-1}) \\
&= a_0 B_0 + \sum_{n=1}^{N} a_n B_n - \sum_{n=0}^{N-1} a_{n+1}B_n \\
&= a_N B_N + \sum_{n=0}^{N-1} a_n B_n - \sum_{n=0}^{N-1} a_{n+1}B_n \\
&= a_N B_N - \sum_{n=0}^{N-1}(a_{n+1} - a_n)B_n.
\end{aligned}
$$

Substituting $n = y_v$, $a_n = f(\boldsymbol{y})$, $b_n = \nabla_{\boldsymbol{\gamma}}q_{\boldsymbol{\gamma}}(y_v)$, and $B_n = \nabla_{\boldsymbol{\gamma}}Q_{\boldsymbol{\gamma}}(y_v)$ into the above equation, we have

$$\nabla_{\boldsymbol{\gamma}}\mathbb{E}_{q_{\boldsymbol{\gamma}}(\boldsymbol{y})}[f(\boldsymbol{y})] =$$

$$\sum_v \mathbb{E}_{q_{\boldsymbol{\gamma}}(\boldsymbol{y}_{-v})}\Big[\underbrace{f(\boldsymbol{y}_{-v}, y_v = N)\nabla_{\boldsymbol{\gamma}}Q_{\boldsymbol{\gamma}}(y_v = N)}_{\text{``0''}} - \underbrace{\sum_{y_v=0}^{N-1}[f(\boldsymbol{y}_{-v}, y_v + 1) - f(\boldsymbol{y})]\nabla_{\boldsymbol{\gamma}}Q_{\boldsymbol{\gamma}}(y_v)}_{\text{``Key''}}\Big].$$

Note the first term equals zero for both finite alphabet, *i.e.,* $N < \infty$, and infinite alphabet, *i.e.,* $N = \infty$. When $N = \infty$, we get (8) in the main manuscript.

With the above proofs for Eqs. (7) and (8), one could straightforwardly verify Theorem 1.

Table 3: Variable-nabla examples. Note $y$ is a scalar random variable.

| $q_\gamma(y)$ | $g_\gamma^{q_\gamma(y)}$ | |
|---|---|---|
| Delta $\delta(y-\mu)$ | $g_\mu^{q_\gamma(y)} = 1$ | |
| Bernoulli$(p)$ | $g_p^{q_\gamma(y)} = \begin{cases} 1/(1-p) & y=0 \\ 0 & y=1 \end{cases}$ | |
| Normal$(\mu,\sigma^2)$ | $g_\mu^{q_\gamma(y)} = 1$ | $g_\sigma^{q_\gamma(y)} = \frac{y-\mu}{\sigma}$ |
| Log-Normal$(\mu,\sigma^2)$ | $g_\mu^{q_\gamma(y)} = y$ | $g_\sigma^{q_\gamma(y)} = y\frac{\log y-\mu}{\sigma}$ |
| Logit-Normal$(\mu,\sigma^2)$ | $g_\mu^{q_\gamma(y)} = y(1-y)$ | $g_\sigma^{q_\gamma(y)} = y(1-y)\frac{\text{logit}(y)-\mu}{\sigma}$ |
| Cauchy$(\mu,\gamma)$ | $g_\mu^{q_\gamma(y)} = 1$ | $g_\gamma^{q_\gamma(y)} = \frac{y-\mu}{\gamma}$ |
| Gamma$\left(\alpha,\frac{1}{\beta}\right)$ | $g_\alpha^{q_\gamma(y)} = \frac{[\log(\beta y)-\psi(\alpha)]\Gamma(\alpha,\beta y)+\beta y T(3,\alpha,\beta y)}{\beta^\alpha y^{\alpha-1}e^{-\beta y}}$ | $g_\beta^{q_\gamma(y)} = -\frac{y}{\beta}$ |
| Beta$(\alpha,\beta)$ | $g_\alpha^{q_\gamma(y)} = \begin{bmatrix} -\frac{\log y - \psi(\alpha)+\psi(\alpha+\beta)}{y^{\alpha-1}(1-y)^{\beta-1}}B(y;\alpha,\beta) \\ +\frac{y}{\alpha^2(1-y)^{\beta-1}}\cdot {}_3F_2(\alpha,\alpha,1-\beta;\alpha+1,\alpha+1;y) \end{bmatrix}$ | |
| | $g_\beta^{q_\gamma(y)} = \begin{bmatrix} \frac{\log(1-y)-\psi(\beta)+\psi(\alpha+\beta)}{y^{\alpha-1}(1-y)^{\beta-1}}B(1-y;\beta,\alpha) \\ -\frac{1-y}{\beta^2 y^{\alpha-1}}\cdot {}_3F_2(\beta,\beta,1-\alpha;\beta+1,\beta+1;1-y) \end{bmatrix}$ | |
| NB$(r,p)$ | $g_r^{q_\gamma(y)} = \begin{bmatrix} -\frac{\log(1-p)-\psi(r)+\psi(r+y+1)}{(1-p)^r p^y}(y+r)B(1-p;r,y+1) \\ +\frac{y+r}{p^y r^2}\cdot {}_3F_2(r,r,-y;r+1,r+1;1-p) \end{bmatrix}$ | |
| | $g_p^{q_\gamma(y)} = \frac{y+r}{1-p}$ | |
| Exponential$(\lambda)$ | $g_\lambda^{q_\gamma(y)} = -\frac{y}{\lambda}$ | |
| Student's t$(v)$ | $g_v^{q_\gamma(y)} = \frac{y}{2}\left(1+\frac{y^2}{v}\right)^{\frac{v+1}{2}}\cdot \begin{bmatrix} \left[\frac{1}{v}+\psi\left(\frac{v}{2}\right)-\psi\left(\frac{v+1}{2}\right)\right]\cdot {}_2F_1\left(\frac{1}{2},\frac{v+1}{2};\frac{3}{2};-\frac{y^2}{v}\right) \\ +\frac{2y(v+1)}{3v}\cdot {}_2F_1\left(\frac{3}{2},\frac{v+3}{2};\frac{5}{2};-\frac{y^2}{v}\right) \\ +\frac{y^2}{3v}\cdot F_{201}^{212}\begin{bmatrix}\frac{3}{2},\frac{v+3}{2};1,1,\frac{v+1}{2}; \\ 2,\frac{5}{2};;\frac{v+3}{2};\end{bmatrix}-\frac{y^2}{v},-\frac{y^2}{v} \end{bmatrix}$ | |
| Weibull$(\lambda,k)$ | $g_\lambda^{q_\gamma(y)} = \frac{y}{\lambda}$ | $g_k^{q_\gamma(y)} = \frac{y}{k}\log\left(\frac{\lambda}{y}\right)$ |
| Laplace$(\mu,b)$ | $g_\mu^{q_\gamma(y)} = 1$ | $g_b^{q_\gamma(y)} = \frac{y-\mu}{b}$ |
| Poisson$(\lambda)$ | $g_\lambda^{q_\gamma(y)} = 1$ | |
| Geometric$(p)$ | $g_p^{q_\gamma(y)} = -\frac{y+1}{p}$ | |
| Categorical$(\boldsymbol{p})$ | $g_{\boldsymbol{p}}^{q_\gamma(y)} = -\frac{1}{p_y}[1_{0\leq y},1_{1\leq y},\cdots,1_{(N-1)\leq y},0]^T$ | |
| | for $y=\{0,1,\cdots,N\}$, $\boldsymbol{p}=[p_0,p_1,\cdots,p_N]^T$ | |

$\psi(x)$ is the digamma function. $\Gamma(x,y)$ is the upper incomplete gamma function. $B(x;\alpha,\beta)$ is the incomplete beta function. ${}_pF_q(a_1,\cdots,a_p;b_1,\cdots,b_p;x)$ is the generalized hypergeometric function. $T(m,s,x)$ is a special case of Meijer G-function (Geddes et al., 1990).

## B   WHY DISCRETE *Internal* VARIABLES ARE CHALLENGING?

For simpler demonstration, we first use a 2-layer model to show why discrete *internal* variables are challenging. Then, we present an importance-sampling proposal that might be useful under specific situations. Finally, we present a strategy to learn discrete *internal* variables with the statistical back-propagation in Theorem 3 of the main manuscript.

Assume $q_\gamma(\boldsymbol{y})$ being the marginal distribution of the following 2-layer model

$$q_\gamma(\boldsymbol{y}) = \mathbb{E}_{q_{\gamma_\lambda}(\boldsymbol{\lambda})}[q_{\gamma_y}(\boldsymbol{y}|\boldsymbol{\lambda})]$$

where $\gamma = \{\gamma_{\boldsymbol{y}},\gamma_{\boldsymbol{\lambda}}\}$, $q_\gamma(\boldsymbol{y},\boldsymbol{\lambda}) = q_{\gamma_y}(\boldsymbol{y}|\boldsymbol{\lambda})q_{\gamma_\lambda}(\boldsymbol{\lambda}) = \prod_v q_{\gamma_y}(y_v|\boldsymbol{\lambda})\cdot\prod_k q_{\gamma_\lambda}(\lambda_k)$, and both the *leaf* variable $\boldsymbol{y}$ and the *internal* variable $\boldsymbol{\lambda}$ could be either continuous or discrete.

Accordingly, the objective becomes

$$\mathbb{E}_{q_\gamma(\boldsymbol{y})}[f(\boldsymbol{y})] = \mathbb{E}_{q_\gamma(\boldsymbol{y},\boldsymbol{\lambda})}[f(\boldsymbol{y})] = \mathbb{E}_{q_{\gamma_y}(\boldsymbol{y}|\boldsymbol{\lambda})q_{\gamma_\lambda}(\boldsymbol{\lambda})}[f(\boldsymbol{y})].$$

For gradient wrt $\boldsymbol{\gamma_y}$, using Theorem 1, it is straight to show

$$\nabla_{\boldsymbol{\gamma_y}} \mathbb{E}_{q_{\boldsymbol{\gamma}}(\boldsymbol{y},\boldsymbol{\lambda})}[f(\boldsymbol{y})] = \mathbb{E}_{q_{\boldsymbol{\gamma_\lambda}}(\boldsymbol{\lambda})}\Big[\nabla_{\boldsymbol{\gamma_y}}\mathbb{E}_{q_{\boldsymbol{\gamma_y}}(\boldsymbol{y}|\boldsymbol{\lambda})}[f(\boldsymbol{y})]\Big] = \mathbb{E}_{q_{\boldsymbol{\gamma}}(\boldsymbol{y},\boldsymbol{\lambda})}\Big[\mathbb{G}_{\boldsymbol{\gamma_y}}^{q_{\boldsymbol{\gamma_y}}(\boldsymbol{y}|\boldsymbol{\lambda})}\mathbb{D}_{\boldsymbol{y}}[f(\boldsymbol{y})]\Big]. \quad (15)$$

For gradient wrt $\boldsymbol{\gamma_\lambda}$, we first have

$$\nabla_{\boldsymbol{\gamma_\lambda}} \mathbb{E}_{q_{\boldsymbol{\gamma}}(\boldsymbol{y},\boldsymbol{\lambda})}[f(\boldsymbol{y})] = \nabla_{\boldsymbol{\gamma_\lambda}} \mathbb{E}_{q_{\boldsymbol{\gamma_\lambda}}(\boldsymbol{\lambda})}\Big[\mathbb{E}_{q_{\boldsymbol{\gamma_y}}(\boldsymbol{y}|\boldsymbol{\lambda})}[f(\boldsymbol{y})]\Big].$$

With $\hat{f}(\boldsymbol{\lambda}) = \mathbb{E}_{q_{\boldsymbol{\gamma_y}}(\boldsymbol{y}|\boldsymbol{\lambda})}[f(\boldsymbol{y})]$, we then apply Theorem 1 and get

$$\nabla_{\boldsymbol{\gamma_\lambda}} \mathbb{E}_{q_{\boldsymbol{\gamma}}(\boldsymbol{y},\boldsymbol{\lambda})}[f(\boldsymbol{y})] = \mathbb{E}_{q_{\boldsymbol{\gamma_\lambda}}(\boldsymbol{\lambda})}\Big[\mathbb{G}_{\boldsymbol{\gamma_\lambda}}^{q_{\boldsymbol{\gamma_\lambda}}(\boldsymbol{\lambda})}\mathbb{D}_{\boldsymbol{\lambda}}[\hat{f}(\boldsymbol{\lambda})]\Big], \quad (16)$$

where $\mathbb{D}_{\boldsymbol{\lambda}}[\hat{f}(\boldsymbol{\lambda})] = [\cdots, \mathbb{D}_{\lambda_k}[\hat{f}(\boldsymbol{\lambda})], \cdots]^T$, and

$$\mathbb{D}_{\lambda_k}[\hat{f}(\boldsymbol{\lambda})] \triangleq \begin{cases} \nabla_{\lambda_k}\hat{f}(\boldsymbol{\lambda}), & \text{Continous } \lambda_k \\ \hat{f}(\boldsymbol{\lambda}_{-k}, \lambda_k + 1) - \hat{f}(\boldsymbol{\lambda}), & \text{Discrete } \lambda_k \end{cases}$$

Next, we separately discuss the situations where $\lambda_k$ is continuous or discrete.

## B.1 For continuous $\lambda_k$

One can directly apply Theorem 1 again, namely

$$\mathbb{D}_{\lambda_k}[\hat{f}(\boldsymbol{\lambda})] = \nabla_{\lambda_k} \mathbb{E}_{q_{\boldsymbol{\gamma_y}}(\boldsymbol{y}|\boldsymbol{\lambda})}[f(\boldsymbol{y})] = \mathbb{E}_{q_{\boldsymbol{\gamma_y}}(\boldsymbol{y}|\boldsymbol{\lambda})}\Big[\mathbb{G}_{\lambda_k}^{q_{\boldsymbol{\gamma_y}}(\boldsymbol{y}|\boldsymbol{\lambda})}\mathbb{D}_{\boldsymbol{y}}[f(\boldsymbol{y})]\Big]. \quad (17)$$

Substituting (17) into (16), we have

$$\nabla_{\boldsymbol{\gamma_\lambda}} \mathbb{E}_{q_{\boldsymbol{\gamma}}(\boldsymbol{y},\boldsymbol{\lambda})}[f(\boldsymbol{y})] = \mathbb{E}_{q_{\boldsymbol{\gamma}}(\boldsymbol{y},\boldsymbol{\lambda})}\Big[\mathbb{G}_{\boldsymbol{\gamma_\lambda}}^{q_{\boldsymbol{\gamma_\lambda}}(\boldsymbol{\lambda})}\mathbb{G}_{\boldsymbol{\lambda}}^{q_{\boldsymbol{\gamma_y}}(\boldsymbol{y}|\boldsymbol{\lambda})}\mathbb{D}_{\boldsymbol{y}}[f(\boldsymbol{y})]\Big]. \quad (18)$$

## B.2 For discrete $\lambda_k$

In this case, we need to calculate $\mathbb{D}_{\lambda_k}[\hat{f}(\boldsymbol{\lambda})] = \hat{f}(\boldsymbol{\lambda}_{-k}, \lambda_k + 1) - \hat{f}(\boldsymbol{\lambda})$. The keys are again partial integration and Abel transformation. For simplicity, we first assume one-dimensional $y$, and separately discuss $y$ being continuous and discrete.

For continuous $y$, we apply partial integration to $\hat{f}(\lambda)$ and get

$$\hat{f}(\boldsymbol{\lambda}) = \mathbb{E}_{q_{\boldsymbol{\gamma_y}}(y|\boldsymbol{\lambda})}[f(y)] = \int q_{\boldsymbol{\gamma_y}}(y|\boldsymbol{\lambda})f(y)dy$$

$$= \left[f(y)Q_{\boldsymbol{\gamma_y}}(y|\boldsymbol{\lambda})\right]\big|_{-\infty}^{\infty} - \int Q_{\boldsymbol{\gamma_y}}(y|\boldsymbol{\lambda})\nabla_y f(y)dy.$$

Accordingly, we have

$$\mathbb{D}_{\lambda_k}[\hat{f}(\boldsymbol{\lambda})] = \hat{f}(\boldsymbol{\lambda}_{-k}, \lambda_k + 1) - \hat{f}(\boldsymbol{\lambda})$$

$$= \int \Big[Q_{\boldsymbol{\gamma_y}}(y|\boldsymbol{\lambda}) - Q_{\boldsymbol{\gamma_y}}(y|\boldsymbol{\lambda}_{-k}, \lambda_k + 1)\Big]\nabla_y f(y)dy$$

$$+ \underbrace{\left[f(y)Q_{\boldsymbol{\gamma_y}}(y|\boldsymbol{\lambda}_{-k}, \lambda_k + 1)\right]\big|_{-\infty}^{\infty} - \left[f(y)Q_{\boldsymbol{\gamma_y}}(y|\boldsymbol{\lambda})\right]\big|_{-\infty}^{\infty}}_{\text{``0''}}.$$

Removing the "0" term, we have

$$\mathbb{D}_{\lambda_k}[\hat{f}(\boldsymbol{\lambda})] = \int \Big[Q_{\boldsymbol{\gamma_y}}(y|\boldsymbol{\lambda}) - Q_{\boldsymbol{\gamma_y}}(y|\boldsymbol{\lambda}_{-k}, \lambda_k + 1)\Big]\nabla_y f(y)dy. \quad (19)$$

For discrete $y$, by similarly exploiting Abel transformation, we have

$$\hat{f}(\boldsymbol{\lambda}) = \mathbb{E}_{q_{\boldsymbol{\gamma_y}}(y|\boldsymbol{\lambda})}[f(y)] = \sum_y q_{\boldsymbol{\gamma_y}}(y|\boldsymbol{\lambda})f(y)$$

$$= f(\infty)Q_{\boldsymbol{\gamma_y}}(\infty|\boldsymbol{\lambda}) - \sum_y Q_{\boldsymbol{\gamma_y}}(y|\boldsymbol{\lambda})\big[f(y+1) - f(y)\big].$$

Accordingly, we get

$$\mathbb{D}_{\lambda_k}[\hat{f}(\boldsymbol{\lambda})] = \sum_y \Big[ Q_{\boldsymbol{\gamma_y}}(y|\boldsymbol{\lambda}) - Q_{\boldsymbol{\gamma_y}}(y|\boldsymbol{\lambda}_{-k}, \lambda_k + 1) \Big] \big[ f(y+1) - f(y) \big]. \tag{20}$$

Unifying (19) for continuous $y$ and (20) for discrete $y$, we have

$$\begin{aligned}
\mathbb{D}_{\lambda_k}[\hat{f}(\boldsymbol{\lambda})] &= \hat{f}(\boldsymbol{\lambda}_{-k}, \lambda_k + 1) - \hat{f}(\boldsymbol{\lambda}) \\
&= \mathbb{E}_{q_{\boldsymbol{\gamma_y}}(y|\boldsymbol{\lambda}_{-k}, \lambda_k+1)}[f(y)] - \mathbb{E}_{q_{\boldsymbol{\gamma_y}}(y|\boldsymbol{\lambda})}[f(y)] \\
&= \mathbb{E}_{q_{\boldsymbol{\gamma_y}}(y|\boldsymbol{\lambda})}\Big[ \mathbb{D}_y[f(y)] \cdot \bar{g}_{\lambda_k}^{q_{\boldsymbol{\gamma_y}}(y|\boldsymbol{\lambda})} \Big]
\end{aligned} \tag{21}$$

where we define

$$\bar{g}_{\lambda_k}^{q_{\boldsymbol{\gamma_y}}(y|\boldsymbol{\lambda})} \triangleq \frac{Q_{\boldsymbol{\gamma_y}}(y|\boldsymbol{\lambda}) - Q_{\boldsymbol{\gamma_y}}(y|\boldsymbol{\lambda}_{-k}, \lambda_k + 1)}{q_{\boldsymbol{\gamma_y}}(y|\boldsymbol{\lambda})}.$$

**Multi-dimensional $y$.** Based on the above one-dimensional foundation, we next move on to multidimensional situations. With definitions $\boldsymbol{y}_{:i} \triangleq \{y_1, \cdots, y_i\}$ and $\boldsymbol{y}_{i:} \triangleq \{y_i, \cdots, y_V\}$, where $V$ is the dimensionality of $\boldsymbol{y}$, we have

$$\begin{aligned}
\mathbb{D}_{\lambda_k}[\hat{f}(\boldsymbol{\lambda})] &= \hat{f}(\boldsymbol{\lambda}_{-k}, \lambda_k + 1) - \hat{f}(\boldsymbol{\lambda}) \\
&= \mathbb{E}_{q_{\boldsymbol{\gamma_y}}(\boldsymbol{y}|\boldsymbol{\lambda}_{-k}, \lambda_k+1)}[f(\boldsymbol{y})] - \mathbb{E}_{q_{\boldsymbol{\gamma_y}}(\boldsymbol{y}|\boldsymbol{\lambda})}[f(\boldsymbol{y})] \\
&= \mathbb{E}_{q_{\boldsymbol{\gamma_y}}(\boldsymbol{y}_{2:}|\boldsymbol{\lambda}_{-k}, \lambda_k+1)}\Big[ \mathbb{E}_{q_{\boldsymbol{\gamma_y}}(y_1|\boldsymbol{\lambda}_{-k}, \lambda_k+1)}[f(\boldsymbol{y})] - \mathbb{E}_{q_{\boldsymbol{\gamma_y}}(y_1|\boldsymbol{\lambda})}[f(\boldsymbol{y})] \\
&\qquad\qquad + \mathbb{E}_{q_{\boldsymbol{\gamma_y}}(y_1|\boldsymbol{\lambda})}[f(\boldsymbol{y})] \Big] - \mathbb{E}_{q_{\boldsymbol{\gamma_y}}(\boldsymbol{y}|\boldsymbol{\lambda})}[f(\boldsymbol{y})]
\end{aligned} \tag{22}$$

Apply (21) and we have

$$\begin{aligned}
\mathbb{D}_{\lambda_k}[\hat{f}(\boldsymbol{\lambda})] &= \mathbb{E}_{q_{\boldsymbol{\gamma_y}}(\boldsymbol{y}_{2:}|\boldsymbol{\lambda}_{-k}, \lambda_k+1)}\Big[ \mathbb{E}_{q_{\boldsymbol{\gamma_y}}(y_1|\boldsymbol{\lambda})}\Big[ \mathbb{D}_{y_1}[f(\boldsymbol{y})] \cdot \bar{g}_{\lambda_k}^{q_{\boldsymbol{\gamma_y}}(y_1|\boldsymbol{\lambda})} \Big] \Big] \\
&\quad + \mathbb{E}_{q_{\boldsymbol{\gamma_y}}(\boldsymbol{y}_{2:}|\boldsymbol{\lambda}_{-k}, \lambda_k+1)}\Big[ \mathbb{E}_{q_{\boldsymbol{\gamma_y}}(y_1|\boldsymbol{\lambda})}[f(\boldsymbol{y})] \Big] - \mathbb{E}_{q_{\boldsymbol{\gamma_y}}(\boldsymbol{y}|\boldsymbol{\lambda})}[f(\boldsymbol{y})] \\
&= \mathbb{E}_{q_{\boldsymbol{\gamma_y}}(\boldsymbol{y}_{2:}|\boldsymbol{\lambda}_{-k}, \lambda_k+1)q_{\boldsymbol{\gamma_y}}(y_1|\boldsymbol{\lambda})}\Big[ \mathbb{D}_{y_1}[f(\boldsymbol{y})] \cdot \bar{g}_{\lambda_k}^{q_{\boldsymbol{\gamma_y}}(y_1|\boldsymbol{\lambda})} + f(\boldsymbol{y}) \Big] - \mathbb{E}_{q_{\boldsymbol{\gamma_y}}(\boldsymbol{y}|\boldsymbol{\lambda})}[f(\boldsymbol{y})]
\end{aligned}$$

Similarly, we add extra terms to the above equation to enable applying (21) again as

$$\begin{aligned}
\mathbb{D}_{\lambda_k}[\hat{f}(\boldsymbol{\lambda})] &= \mathbb{E}_{q_{\boldsymbol{\gamma_y}}(\boldsymbol{y}_{2:}|\boldsymbol{\lambda}_{-k}, \lambda_k+1)q_{\boldsymbol{\gamma_y}}(y_1|\boldsymbol{\lambda})}\Big[ \mathbb{D}_{y_1}[f(\boldsymbol{y})] \cdot \bar{g}_{\lambda_k}^{q_{\boldsymbol{\gamma_y}}(y_1|\boldsymbol{\lambda})} + f(\boldsymbol{y}) \Big] \\
&\quad - \mathbb{E}_{q_{\boldsymbol{\gamma_y}}(\boldsymbol{y}_{3:}|\boldsymbol{\lambda}_{-k}, \lambda_k+1)q_{\boldsymbol{\gamma_y}}(\boldsymbol{y}_{:2}|\boldsymbol{\lambda})}\Big[ \mathbb{D}_{y_1}[f(\boldsymbol{y})] \cdot \bar{g}_{\lambda_k}^{q_{\boldsymbol{\gamma_y}}(y_1|\boldsymbol{\lambda})} + f(\boldsymbol{y}) \Big] \\
&\quad + \mathbb{E}_{q_{\boldsymbol{\gamma_y}}(\boldsymbol{y}_{3:}|\boldsymbol{\lambda}_{-k}, \lambda_k+1)q_{\boldsymbol{\gamma_y}}(\boldsymbol{y}_{:2}|\boldsymbol{\lambda})}\Big[ \mathbb{D}_{y_1}[f(\boldsymbol{y})] \cdot \bar{g}_{\lambda_k}^{q_{\boldsymbol{\gamma_y}}(y_1|\boldsymbol{\lambda})} + f(\boldsymbol{y}) \Big] \\
&\quad - \mathbb{E}_{q_{\boldsymbol{\gamma_y}}(\boldsymbol{y}|\boldsymbol{\lambda})}[f(\boldsymbol{y})].
\end{aligned}$$

Accordingly, we apply (21) to the first two terms and have

$$\begin{aligned}
\mathbb{D}_{\lambda_k}[\hat{f}(\boldsymbol{\lambda})] &= \mathbb{E}_{q_{\boldsymbol{\gamma_y}}(\boldsymbol{y}_{3:}|\boldsymbol{\lambda}_{-k}, \lambda_k+1)q_{\boldsymbol{\gamma_y}}(\boldsymbol{y}_{:2}|\boldsymbol{\lambda})}\Big[ \mathbb{D}_{y_2}\Big[ \mathbb{D}_{y_1}[f(\boldsymbol{y})]\bar{g}_{\lambda_k}^{q_{\boldsymbol{\gamma_y}}(y_1|\boldsymbol{\lambda})} + f(\boldsymbol{y}) \Big] \bar{g}_{\lambda_k}^{q_{\boldsymbol{\gamma_y}}(y_2|\boldsymbol{\lambda})} \Big] \\
&\quad + \mathbb{E}_{q_{\boldsymbol{\gamma_y}}(\boldsymbol{y}_{3:}|\boldsymbol{\lambda}_{-k}, \lambda_k+1)q_{\boldsymbol{\gamma_y}}(\boldsymbol{y}_{:2}|\boldsymbol{\lambda})}\Big[ \mathbb{D}_{y_1}[f(\boldsymbol{y})]\bar{g}_{\lambda_k}^{q_{\boldsymbol{\gamma_y}}(y_1|\boldsymbol{\lambda})} + f(\boldsymbol{y}) \Big] - \mathbb{E}_{q_{\boldsymbol{\gamma_y}}(\boldsymbol{y}|\boldsymbol{\lambda})}[f(\boldsymbol{y})] \\
&= \mathbb{E}_{q_{\boldsymbol{\gamma_y}}(\boldsymbol{y}_{3:}|\boldsymbol{\lambda}_{-k}, \lambda_k+1)q_{\boldsymbol{\gamma_y}}(\boldsymbol{y}_{:2}|\boldsymbol{\lambda})}\left[ \begin{array}{l} \mathbb{D}_{y_2}\Big[ \mathbb{D}_{y_1}[f(\boldsymbol{y})]\bar{g}_{\lambda_k}^{q_{\boldsymbol{\gamma_y}}(y_1|\boldsymbol{\lambda})} + f(\boldsymbol{y}) \Big] \bar{g}_{\lambda_k}^{q_{\boldsymbol{\gamma_y}}(y_2|\boldsymbol{\lambda})} \\ + \mathbb{D}_{y_1}[f(\boldsymbol{y})]\bar{g}_{\lambda_k}^{q_{\boldsymbol{\gamma_y}}(y_1|\boldsymbol{\lambda})} + f(\boldsymbol{y}) \end{array} \right] \\
&\quad - \mathbb{E}_{q_{\boldsymbol{\gamma_y}}(\boldsymbol{y}|\boldsymbol{\lambda})}[f(\boldsymbol{y})]
\end{aligned}$$

So forth, we summarize the pattern into the following equation as

$$\mathbb{D}_{\lambda_k}[\hat{f}(\boldsymbol{\lambda})] = \mathbb{E}_{q_{\boldsymbol{\gamma_y}}(\boldsymbol{y}|\boldsymbol{\lambda})}\big[ \mathbb{A}_{\lambda_k}(\boldsymbol{y}, V) - f(\boldsymbol{y}) \big], \tag{23}$$

where $\mathbb{A}_{\lambda_k}(\boldsymbol{y}, V)$ is iteratively calculated as

$$\mathbb{A}_{\lambda_k}(\boldsymbol{y}, 0) = f(\boldsymbol{y})$$

$$\mathbb{A}_{\lambda_k}(\boldsymbol{y}, v) = \mathbb{D}_{y_v}\big[\mathbb{A}_{\lambda_k}(\boldsymbol{y}, v-1)\big]\bar{g}_{\lambda_k}^{q_{\gamma_y}(y_v|\boldsymbol{\lambda})} + \mathbb{A}_{\lambda_k}(\boldsymbol{y}, v-1)$$

$$\mathbb{A}_{\lambda_k}(\boldsymbol{y}, V) = \mathbb{D}_{y_V}\big[\mathbb{A}_{\lambda_k}(\boldsymbol{y}, V-1)\big]\bar{g}_{\lambda_k}^{q_{\gamma_y}(y_V|\boldsymbol{\lambda})} + \mathbb{A}_{\lambda_k}(\boldsymbol{y}, V-1).$$

Despite elegant structures within (23), to calculate it, one must iterate over all dimensions of $\boldsymbol{y}$, which is computational expensive in practice. More importantly, it is straightforward to show that deeper models will have similar but much more complicated expressions.

### B.3   An Importance-sampling Proposal to Handle Discrete *Internal* Variables

Next, we present an intuitive proposal that might be useful under specific situations.

The key idea is to use different extra items to enable "easy-to-use" expression for $\mathbb{D}_{\lambda_k}[\hat{f}(\boldsymbol{\lambda})]$, namely

$$\begin{aligned}
\mathbb{D}_{\lambda_k}[\hat{f}(\boldsymbol{\lambda})] &= \hat{f}(\boldsymbol{\lambda}_{-k}, \lambda_k + 1) - \hat{f}(\boldsymbol{\lambda}) \\
&= \mathbb{E}_{q_{\gamma_y}(\boldsymbol{y}|\boldsymbol{\lambda}_{-k}, \lambda_k+1)}[f(\boldsymbol{y})] - \mathbb{E}_{q_{\gamma_y}(\boldsymbol{y}|\boldsymbol{\lambda})}[f(\boldsymbol{y})] \\
&= \mathbb{E}_{q_{\gamma_y}(\boldsymbol{y}|\boldsymbol{\lambda}_{-k}, \lambda_k+1)}[f(\boldsymbol{y})] \\
&\quad - \mathbb{E}_{q_{\gamma_y}(\boldsymbol{y}_{:V-1}|\boldsymbol{\lambda}_{-k}, \lambda_k+1)q_{\gamma_y}(y_V|\boldsymbol{\lambda})}[f(\boldsymbol{y})] \\
&\quad + \mathbb{E}_{q_{\gamma_y}(\boldsymbol{y}_{:V-1}|\boldsymbol{\lambda}_{-k}, \lambda_k+1)q_{\gamma_y}(y_V|\boldsymbol{\lambda})}[f(\boldsymbol{y})] \\
&\quad \cdots \\
&\quad - \mathbb{E}_{q_{\gamma_y}(\boldsymbol{y}_{:i-1}|\boldsymbol{\lambda}_{-k}, \lambda_k+1)q_{\gamma_y}(\boldsymbol{y}_{i:}|\boldsymbol{\lambda})}[f(\boldsymbol{y})] \\
&\quad + \mathbb{E}_{q_{\gamma_y}(\boldsymbol{y}_{:i-1}|\boldsymbol{\lambda}_{-k}, \lambda_k+1)q_{\gamma_y}(\boldsymbol{y}_{i:}|\boldsymbol{\lambda})}[f(\boldsymbol{y})] \\
&\quad \cdots \\
&\quad - \mathbb{E}_{q_{\gamma_y}(y_1|\boldsymbol{\lambda}_{-k}, \lambda_k+1)q_{\gamma_y}(\boldsymbol{y}_{2:}|\boldsymbol{\lambda})}[f(\boldsymbol{y})] \\
&\quad + \mathbb{E}_{q_{\gamma_y}(y_1|\boldsymbol{\lambda}_{-k}, \lambda_k+1)q_{\gamma_y}(\boldsymbol{y}_{2:}|\boldsymbol{\lambda})}[f(\boldsymbol{y})] \\
&\quad - \mathbb{E}_{q_{\gamma_y}(\boldsymbol{y}|\boldsymbol{\lambda})}[f(\boldsymbol{y})].
\end{aligned}$$

Apply (21) to the adjacent two terms for $V$ times, we have

$$\begin{aligned}
\mathbb{D}_{\lambda_k}[\hat{f}(\boldsymbol{\lambda})] &= \mathbb{E}_{q_{\gamma_y}(\boldsymbol{y}_{:V-1}|\boldsymbol{\lambda}_{-k}, \lambda_k+1)q_{\gamma_y}(y_V|\boldsymbol{\lambda})}\Big[\mathbb{D}_{y_V}[f(\boldsymbol{y})]\bar{g}_{\lambda_k}^{q_{\gamma_y}(y_V|\boldsymbol{\lambda})}\Big] \\
&\quad \cdots \\
&\quad + \mathbb{E}_{q_{\gamma_y}(\boldsymbol{y}_{:i-1}|\boldsymbol{\lambda}_{-k}, \lambda_k+1)q_{\gamma_y}(\boldsymbol{y}_{i:}|\boldsymbol{\lambda})}\Big[\mathbb{D}_{y_i}[f(\boldsymbol{y})]\bar{g}_{\lambda_k}^{q_{\gamma_y}(y_i|\boldsymbol{\lambda})}\Big] \\
&\quad \cdots \\
&\quad + \mathbb{E}_{q_{\gamma_y}(y_V|\boldsymbol{\lambda})}\Big[\mathbb{D}_{y_1}[f(\boldsymbol{y})]\bar{g}_{\lambda_k}^{q_{\gamma_y}(y_1|\boldsymbol{\lambda})}\Big],
\end{aligned}$$

where we can apply the idea of importance sampling and modify the above equation to

$$\begin{aligned}
\mathbb{D}_{\lambda_k}[\hat{f}(\boldsymbol{\lambda})] &= \mathbb{E}_{q_{\gamma_y}(\boldsymbol{y}|\boldsymbol{\lambda})}\Big[\mathbb{D}_{y_V}[f(\boldsymbol{y})]\bar{g}_{\lambda_k}^{q_{\gamma_y}(y_V|\boldsymbol{\lambda})}\frac{q_{\gamma_y}(\boldsymbol{y}_{:V-1}|\boldsymbol{\lambda}_{-k}, \lambda_k+1)}{q_{\gamma_y}(\boldsymbol{y}_{:V-1}|\boldsymbol{\lambda})}\Big] \\
&\quad \cdots \\
&\quad + \mathbb{E}_{q_{\gamma_y}(\boldsymbol{y}|\boldsymbol{\lambda})}\Big[\mathbb{D}_{y_i}[f(\boldsymbol{y})]\bar{g}_{\lambda_k}^{q_{\gamma_y}(y_i|\boldsymbol{\lambda})}\frac{q_{\gamma_y}(\boldsymbol{y}_{:i-1}|\boldsymbol{\lambda}_{-k}, \lambda_k+1)}{q_{\gamma_y}(\boldsymbol{y}_{:i-1}|\boldsymbol{\lambda})}\Big] \\
&\quad \cdots \\
&\quad + \mathbb{E}_{q_{\gamma_y}(y_V|\boldsymbol{\lambda})}\Big[\mathbb{D}_{y_1}[f(\boldsymbol{y})]\bar{g}_{\lambda_k}^{q_{\gamma_y}(y_1|\boldsymbol{\lambda})}\Big] \\
&= \mathbb{E}_{q_{\gamma_y}(\boldsymbol{y}|\boldsymbol{\lambda})}\Big[\sum_{v=1}^{V}\mathbb{D}_{y_v}[f(\boldsymbol{y})]\bar{g}_{\lambda_k}^{q_{\gamma_y}(y_v|\boldsymbol{\lambda})}\frac{q_{\gamma_y}(\boldsymbol{y}_{:v-1}|\boldsymbol{\lambda}_{-k}, \lambda_k+1)}{q_{\gamma_y}(\boldsymbol{y}_{:v-1}|\boldsymbol{\lambda})}\Big].
\end{aligned} \quad (24)$$

Note that importance sampling may not always work well in practice (Bishop, 2006).

We further define the **generalized variable-nabla** as

$$
gg_{\lambda_k}^{q_{\gamma_y}(y_v|\boldsymbol{\lambda})} \triangleq
\begin{cases}
\dfrac{-1}{q_{\gamma_y}(y_v|\boldsymbol{\lambda})} \nabla_{\lambda_k} Q_{\gamma_y}(y_v|\boldsymbol{\lambda}), & \text{Continuous } \lambda_k \\[2ex]
\dfrac{Q_{\gamma_y}(y_v|\boldsymbol{\lambda}) - Q_{\gamma_y}(y_v|\boldsymbol{\lambda}_{-k}, \lambda_k + 1)}{q_{\gamma_y}(y_v|\boldsymbol{\lambda})} \dfrac{q_{\gamma_y}(\boldsymbol{y}_{:v-1}|\boldsymbol{\lambda}_{-k}, \lambda_k + 1)}{q_{\gamma_y}(\boldsymbol{y}_{:v-1}|\boldsymbol{\lambda})}, & \text{Discrete } \lambda_k
\end{cases}
\tag{25}
$$

With the generalized variable-nabla, we unify (24) for discrete $\lambda_k$ and (17) for continuous $\lambda_k$ and get

$$
\mathbb{D}_{\lambda_k}[\hat{f}(\boldsymbol{\lambda})] = \mathbb{E}_{q_{\gamma_y}(\boldsymbol{y}|\boldsymbol{\lambda})}\Big[\sum_v \mathbb{D}_{y_v}[f(\boldsymbol{y})] \cdot gg_{\lambda_k}^{q_\gamma(y_v|\boldsymbol{\lambda})}\Big],
$$

which apparently obeys the chain rule. Accordingly, we have the gradient for $\boldsymbol{\gamma_\lambda}$ in (16) as

$$
\nabla_{\boldsymbol{\gamma_\lambda}} \mathbb{E}_{q_\gamma(\boldsymbol{y}, \boldsymbol{\lambda})}[f(\boldsymbol{y})] = \mathbb{E}_{q_\gamma(\boldsymbol{y}, \boldsymbol{\lambda})}\bigg[\sum_k \Big[\sum_v \mathbb{D}_{y_v}[f(\boldsymbol{y})] \cdot gg_{\lambda_k}^{q_\gamma(y_v|\boldsymbol{\lambda})}\Big] \cdot gg_{\boldsymbol{\gamma_\lambda}}^{q_{\gamma_\lambda}(\lambda_k)}\bigg].
$$

One can straightforwardly verify that, with the generalized variable-nabla defined in (25), the chain rule applies to $\nabla_\gamma \mathbb{E}_{q_\gamma(\boldsymbol{y}^{(L)})}[f(\boldsymbol{y}^{(L)})]$, where one can freely specify both *leaf* and *internal* variables to be either continuous or discrete. The only problem is that, for discrete *internal* variables, the importance sampling trick used in (24) may not always work as expected.

## B.4 Strategy to Learn Discrete Internal Variables with Statistical Back-Propagation

Practically, if one has to deal with a $q_\gamma(\boldsymbol{y}^{(1)}, \cdots, \boldsymbol{y}^{(L)})$ with discrete *internal* variables $\boldsymbol{y}^{(l)}, l < L$, we suggest the strategy in Figure 7, with which one should expect a close performance but enjoy much easier implementation with statistical back-propagation in Theorem 3 of the main manuscript. In fact, one can always add additional continuous internal variables to the graphical models to remedy the performance loss or even boost the performance.

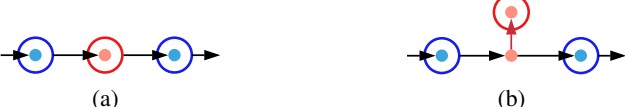

Figure 7: A strategy for discrete *internal* variables. Blue and red circles denote continuous and discrete variables, respectively. The centered dots represent the corresponding distribution parameters. (a) Practically, one uses a neural network (black arrow) to connect the left variable to the parameters of the center discrete one, and then uses another neural network to propagate the sampled value to the next. (b) Instead, we suggest "extracting" the discrete variable as a *leaf* one and propagate its parameters to the next.

## C Proof of Lemma 1

First, a marginal distribution $q_\gamma(\boldsymbol{y})$ with reparameterization $\boldsymbol{y} = \boldsymbol{\tau}_\gamma(\boldsymbol{\epsilon})$, $\boldsymbol{\epsilon} \sim q(\boldsymbol{\epsilon})$ can be expressed as a joint distribution, namely

$$
q_\gamma(\boldsymbol{y}) = q_\gamma(\boldsymbol{y}, \boldsymbol{\epsilon}) = q_\gamma(\boldsymbol{y}|\boldsymbol{\epsilon})q(\boldsymbol{\epsilon}),
$$

where $q_\gamma(\boldsymbol{y}|\boldsymbol{\epsilon}) = \delta(\boldsymbol{y} - \boldsymbol{\tau}_\gamma(\boldsymbol{\epsilon}))$, $\delta(\cdot)$ is the Dirac delta function, and $\boldsymbol{\tau}_\gamma(\boldsymbol{\epsilon})$ could be flexibly specified as a injective, surjective, or bijective function.

Next, we align notations and rewrite (10) as

$$
\begin{aligned}
\nabla_\gamma \mathbb{E}_{q_\gamma(\boldsymbol{y})}[f(\boldsymbol{y})] &= \mathbb{E}_{q_\gamma(\boldsymbol{y}, \boldsymbol{\epsilon})}\Big[\mathbb{G}_\gamma^{q_\gamma(\boldsymbol{y}|\boldsymbol{\epsilon})} \mathbb{D}_{\boldsymbol{y}}[f(\boldsymbol{y})]\Big] \\
&= \mathbb{E}_{q(\boldsymbol{\epsilon})}\Big[\mathbb{E}_{q_\gamma(\boldsymbol{y}|\boldsymbol{\epsilon})}\big[\mathbb{G}_\gamma^{q_\gamma(\boldsymbol{y}|\boldsymbol{\epsilon})} \mathbb{D}_{\boldsymbol{y}}[f(\boldsymbol{y})]\big]\Big],
\end{aligned}
\tag{26}
$$

where $\mathbb{G}_\gamma^{q_\gamma(\boldsymbol{y}|\boldsymbol{\epsilon})} = \big[\cdots, g_\gamma^{q_\gamma(y_v|\boldsymbol{\epsilon})}, \cdots\big]$ and $g_\gamma^{q_\gamma(y_v|\boldsymbol{\epsilon})} \triangleq \frac{-1}{q_\gamma(y_v|\boldsymbol{\epsilon})}\nabla_\gamma Q_\gamma(y_v|\boldsymbol{\epsilon})$.

With $q_{\boldsymbol{\gamma}}(y_v|\boldsymbol{\epsilon}) = \delta(y_v - [\boldsymbol{\tau}_{\boldsymbol{\gamma}}(\boldsymbol{\epsilon})]_v)$, we have

$$Q_{\boldsymbol{\gamma}}(y_v|\boldsymbol{\epsilon}) = \begin{cases} 1 & [\boldsymbol{\tau}_{\boldsymbol{\gamma}}(\boldsymbol{\epsilon})]_v \le y_v \\ 0 & [\boldsymbol{\tau}_{\boldsymbol{\gamma}}(\boldsymbol{\epsilon})]_v > y_v \end{cases}$$

Accordingly, we have

$$\begin{aligned} \nabla_{\boldsymbol{\gamma}} Q_{\boldsymbol{\gamma}}(y_v|\boldsymbol{\epsilon}) &= \nabla_{[\boldsymbol{\tau}_{\boldsymbol{\gamma}}(\boldsymbol{\epsilon})]_v} Q_{\boldsymbol{\gamma}}(y_v|\boldsymbol{\epsilon}) \cdot \nabla_{\boldsymbol{\gamma}}[\boldsymbol{\tau}_{\boldsymbol{\gamma}}(\boldsymbol{\epsilon})]_v \\ &= -\delta([\boldsymbol{\tau}_{\boldsymbol{\gamma}}(\boldsymbol{\epsilon})]_v - y_v) \cdot \nabla_{\boldsymbol{\gamma}}[\boldsymbol{\tau}_{\boldsymbol{\gamma}}(\boldsymbol{\epsilon})]_v \end{aligned}$$

and

$$g_{\boldsymbol{\gamma}}^{q_{\boldsymbol{\gamma}}(y_v|\boldsymbol{\epsilon})} = \frac{-1}{q_{\boldsymbol{\gamma}}(y_v|\boldsymbol{\epsilon})} \nabla_{\boldsymbol{\gamma}} Q_{\boldsymbol{\gamma}}(y_v|\boldsymbol{\epsilon}) = \nabla_{\boldsymbol{\gamma}}[\boldsymbol{\tau}_{\boldsymbol{\gamma}}(\boldsymbol{\epsilon})]_v.$$

Substituting the above equations into (26), we get

$$\nabla_{\boldsymbol{\gamma}} \mathbb{E}_{q_{\boldsymbol{\gamma}}(\boldsymbol{y})}[f(\boldsymbol{y})] = \mathbb{E}_{q(\boldsymbol{\epsilon})}\Big[[\nabla_{\boldsymbol{\gamma}} \boldsymbol{\tau}_{\boldsymbol{\gamma}}(\boldsymbol{\epsilon})][\nabla_{\boldsymbol{y}} f(\boldsymbol{y})]|_{\boldsymbol{y}=\boldsymbol{\tau}_{\boldsymbol{\gamma}}(\boldsymbol{\epsilon})}\Big], \tag{27}$$

which is the multi-dimensional Rep gradient in (5) of the main manuscript.

## D    PROOF OF THEOREM 2

Firstly, with the internal variable $\boldsymbol{\lambda}$ being continuous, (10) and (11) in the main manuscript are proved by (15) and (18) in Section B, respectively. Then, by iteratively generalizing the similar derivations to deep models and utilizing the fact that the GO gradients with variable-nablas in expectation obey the chain rule for models with continuous *internal* variables, Theorem 2 could be readily verified.

## E    PROOFS FOR COROLLARY 1

When all $q_{\boldsymbol{\gamma}^{(i)}}\big(\boldsymbol{y}^{(i)}|\boldsymbol{y}^{(i-1)}\big)$s are specified as Dirac delta functions, namely

$$q_{\boldsymbol{\gamma}^{(i)}}\big(\boldsymbol{y}^{(i)}|\boldsymbol{y}^{(i-1)}\big) = \delta(\boldsymbol{y}^{(i)} - \boldsymbol{\sigma}(\boldsymbol{\gamma}^{(i)}, \boldsymbol{y}^{(i-1)})),$$

where $\boldsymbol{\sigma}(\boldsymbol{\gamma}^{(i)}, \boldsymbol{y}^{(i-1)})$ denotes the activated values after activation functions, the objective becomes

$$\begin{aligned} \mathbb{E}_{q_{\boldsymbol{\gamma}}(\boldsymbol{y}^{(L)})}[f(\boldsymbol{y}^{(L)})] &= f(\boldsymbol{y}^{(L)}) = f(\boldsymbol{\sigma}(\boldsymbol{\gamma}^{(L)}, \boldsymbol{y}^{(L-1)})) \\ &= f(\boldsymbol{\sigma}(\boldsymbol{\gamma}^{(L)}, \boldsymbol{\sigma}(\boldsymbol{\gamma}^{(L-1)}, \boldsymbol{y}^{(L-1)}))) \\ &= f(\boldsymbol{\sigma}(\boldsymbol{\gamma}^{(L)}, \boldsymbol{\sigma}(\boldsymbol{\gamma}^{(L-1)}, \cdots, \boldsymbol{\sigma}(\boldsymbol{\gamma}^{(1)})))) \\ &= f(\boldsymbol{\gamma}), \end{aligned} \tag{28}$$

where $\boldsymbol{\gamma} = \{\boldsymbol{\gamma}^{(1)}, \cdots, \boldsymbol{\gamma}^{(N)}\}$.

**Back-Propagation.** For the objective in (28), the Back-Propagation is expressed as

$$\nabla_{\boldsymbol{\gamma}^{(i)}} f(\boldsymbol{\gamma}) = [\nabla_{\boldsymbol{\gamma}^{(i)}} \boldsymbol{y}^{(i)}][\nabla_{\boldsymbol{y}^{(i)}} f(\cdot)], \tag{29}$$

where

$$\nabla_{\boldsymbol{y}^{(i)}} f(\cdot) = [\nabla_{\boldsymbol{y}^{(i)}} \boldsymbol{y}^{(i+1)}][\nabla_{\boldsymbol{y}^{(i+1)}} f(\cdot)].$$

**Deep GO Gradient.**    We consider the continuous special case, where $\mathbb{D}_{\boldsymbol{y}^{(L)}}\big[f(\boldsymbol{y}^{(L)})\big] = \nabla_{\boldsymbol{y}^{(L)}} f(\boldsymbol{y}^{(L)})$.

With $q_{\boldsymbol{\gamma}^{(i+1)}}\big(\boldsymbol{y}^{(i+1)}|\boldsymbol{y}^{(i)}\big)$s being Dirac delta functions, namely,

$$q_{\boldsymbol{\gamma}^{(i+1)}}\big(y_k^{(i+1)}|\boldsymbol{y}^{(i)}\big) = \begin{cases} \infty, & [\boldsymbol{\sigma}(\boldsymbol{\gamma}^{(i+1)}, \boldsymbol{y}^{(i)})]_k = y_k^{(i+1)} \\ 0, & [\boldsymbol{\sigma}(\boldsymbol{\gamma}^{(i+1)}, \boldsymbol{y}^{(i)})]_k \neq y_k^{(i+1)} \end{cases}$$

we have

$$Q_{\boldsymbol{\gamma}^{(i+1)}}\big(y_k^{(i+1)}|\boldsymbol{y}^{(i)}\big) = \begin{cases} 1, & [\boldsymbol{\sigma}(\boldsymbol{\gamma}^{(i+1)}, \boldsymbol{y}^{(i)})]_k \le y_k^{(i+1)} \\ 0, & [\boldsymbol{\sigma}(\boldsymbol{\gamma}^{(i+1)}, \boldsymbol{y}^{(i)})]_k > y_k^{(i+1)} \end{cases}$$

Taking derivative wrt $y_v^{(i)}$, we got

$$\nabla_{y_v^{(i)}} Q_{\gamma^{(i+1)}}\left(y_k^{(i+1)}|\boldsymbol{y}^{(i)}\right)$$

$$= \nabla_{[\boldsymbol{\sigma}(\boldsymbol{\gamma}^{(i+1)},\boldsymbol{y}^{(i)})]_k} Q_{\gamma^{(i+1)}}\left(y_k^{(i+1)}|\boldsymbol{y}^{(i)}\right) \cdot \nabla_{y_v^{(i)}}[\boldsymbol{\sigma}(\boldsymbol{\gamma}^{(i+1)},\boldsymbol{y}^{(i)})]_k$$

$$= -\delta\left([\boldsymbol{\sigma}(\boldsymbol{\gamma}^{(i+1)},\boldsymbol{y}^{(i)})]_k - y_k^{(i+1)}\right) \cdot \nabla_{y_v^{(i)}}[\boldsymbol{\sigma}(\boldsymbol{\gamma}^{(i+1)},\boldsymbol{y}^{(i)})]_k$$

Accordingly, we have

$$g_{y_v^{(i)}}^{q_{\gamma^{(i+1)}}(y_k^{(i+1)}|\boldsymbol{y}^{(i)})} = \frac{-1}{q_{\gamma^{(i+1)}}(y_k^{(i+1)}|\boldsymbol{y}^{(i)})}\nabla_{y_v^{(i)}} Q_{\gamma^{(i+1)}}(y_k^{(i+1)}|\boldsymbol{y}^{(i)})$$

$$= \nabla_{y_v^{(i)}}[\boldsymbol{\sigma}(\boldsymbol{\gamma}^{(i+1)},\boldsymbol{y}^{(i)})]_k = \nabla_{y_v^{(i)}} y_k^{(i+1)}$$

By substituting the above equation into (12) in Theorem 2, and then comparing it with(29), one can easily verify Corollary 1.

## F  PROOF FOR THEOREM 3

Based on the proofs for Theorem 1 and Theorem 2, it is clear that, if one constrains all *internal* variables to be continuous, the GO gradients in expectation obey the chain rule. Therefore, one can straightforwardly utilizing the chain rule to verify Theorem 3. Actually, Theorem 3 may be seen as the chain rule generalized with random variables, among which the *internal* ones are only allowed to be continuous.

## G  DERIVATIONS FOR HIERARCHICAL VARIATIONAL INFERENCE

In Hierarchical Variational Inference, the objective is to maximize the evidence lower Bound (ELBO)

$$\text{ELBO}(\boldsymbol{\theta},\boldsymbol{\phi};\boldsymbol{x}) = \mathbb{E}_{q_{\boldsymbol{\phi}}(\boldsymbol{z}|\boldsymbol{x})}[\log p_{\boldsymbol{\theta}}(\boldsymbol{x},\boldsymbol{z}) - \log q_{\boldsymbol{\phi}}(\boldsymbol{z}|\boldsymbol{x})]. \tag{30}$$

For the common case with $\boldsymbol{z} = \{\boldsymbol{z}^{(1)},\cdots,\boldsymbol{z}^{(L)}\}$, it is obvious that Theorem 3 of the main manuscript can be applied when optimizing $\boldsymbol{\phi}$.

Practically, there are situations where one might further put a latent variable $\boldsymbol{\lambda}$ in reference $q_{\boldsymbol{\phi}}(\boldsymbol{z}|\boldsymbol{x})$, namely $q_{\boldsymbol{\phi}}(\boldsymbol{z}|\boldsymbol{x}) = \int q_{\boldsymbol{\phi}_z}(\boldsymbol{z}|\boldsymbol{\lambda})q_{\boldsymbol{\phi}_{\boldsymbol{\lambda}}}(\boldsymbol{\lambda})d\boldsymbol{\lambda}$ with $\boldsymbol{\phi} = \{\boldsymbol{\phi}_z,\boldsymbol{\phi}_{\boldsymbol{\lambda}}\}$. Following Ranganath et al. (2016b), we briefly discuss this situation here.

We first show that there is another unnecessary variance-injecting "0" term.

$$\nabla_{\boldsymbol{\phi}}\text{ELBO}(\boldsymbol{\theta},\boldsymbol{\phi};\boldsymbol{x}) = \int [\nabla_{\boldsymbol{\phi}}q_{\boldsymbol{\phi}}(\boldsymbol{z}|\boldsymbol{x})][\log p_{\boldsymbol{\theta}}(\boldsymbol{x},\boldsymbol{z}) - \log q_{\boldsymbol{\phi}}(\boldsymbol{z}|\boldsymbol{x})]d\boldsymbol{z}$$

$$\underbrace{- \int q_{\boldsymbol{\phi}}(\boldsymbol{z}|\boldsymbol{x})\nabla_{\boldsymbol{\phi}}\log q_{\boldsymbol{\phi}}(\boldsymbol{z}|\boldsymbol{x})d\boldsymbol{z}}_{\text{"0"}}, \tag{31}$$

where the second "0" term is straightly verified as

$$\int q_{\boldsymbol{\phi}}(\boldsymbol{z}|\boldsymbol{x})\nabla_{\boldsymbol{\phi}}\log q_{\boldsymbol{\phi}}(\boldsymbol{z}|\boldsymbol{x})d\boldsymbol{z} = \int \nabla_{\boldsymbol{\phi}}q_{\boldsymbol{\phi}}(\boldsymbol{z}|\boldsymbol{x})d\boldsymbol{z} = \nabla_{\boldsymbol{\phi}}\int q_{\boldsymbol{\phi}}(\boldsymbol{z}|\boldsymbol{x})d\boldsymbol{z} = \nabla_{\boldsymbol{\phi}}1 = 0.$$

Eliminating the "0" term from (31), one still has another problem, that is, $\log q_{\boldsymbol{\phi}}(\boldsymbol{z}|\boldsymbol{x})$ is usually non-trivial when $q_{\boldsymbol{\phi}}(\boldsymbol{z}|\boldsymbol{x})$ is marginal. For this problem, we follow Ranganath et al. (2016b) to use another lower bound *ELBO2* of the ELBO in (30).

$$-\log q_{\boldsymbol{\phi}}(\boldsymbol{z}|\boldsymbol{x}) = \int q_{\boldsymbol{\phi}}(\boldsymbol{\lambda}|\boldsymbol{z},\boldsymbol{x})[-\log q_{\boldsymbol{\phi}}(\boldsymbol{z}|\boldsymbol{x})]d\boldsymbol{\lambda} = \int q_{\boldsymbol{\phi}}(\boldsymbol{\lambda}|\boldsymbol{z},\boldsymbol{x})\left[-\log\frac{q_{\boldsymbol{\phi}}(\boldsymbol{z},\boldsymbol{\lambda}|\boldsymbol{x})}{q_{\boldsymbol{\phi}}(\boldsymbol{\lambda}|\boldsymbol{z},\boldsymbol{x})}\right]d\boldsymbol{\lambda}$$

$$= \mathbb{E}_{q_{\boldsymbol{\phi}}(\boldsymbol{\lambda}|\boldsymbol{z},\boldsymbol{x})}\left[-\log\frac{q_{\boldsymbol{\phi}}(\boldsymbol{z},\boldsymbol{\lambda}|\boldsymbol{x})}{r_{\boldsymbol{\omega}}(\boldsymbol{\lambda}|\boldsymbol{z},\boldsymbol{x})} + \log\frac{q_{\boldsymbol{\phi}}(\boldsymbol{\lambda}|\boldsymbol{z},\boldsymbol{x})}{r_{\boldsymbol{\omega}}(\boldsymbol{\lambda}|\boldsymbol{z},\boldsymbol{x})}\right]$$

$$\geq \mathbb{E}_{q_{\boldsymbol{\phi}}(\boldsymbol{\lambda}|\boldsymbol{z},\boldsymbol{x})}\left[-\log\frac{q_{\boldsymbol{\phi}}(\boldsymbol{z},\boldsymbol{\lambda}|\boldsymbol{x})}{r_{\boldsymbol{\omega}}(\boldsymbol{\lambda}|\boldsymbol{z},\boldsymbol{x})}\right]$$

where $r_{\boldsymbol{\omega}}(\boldsymbol{\lambda}|\boldsymbol{z}, \boldsymbol{x})$, evaluable, is an additional variational distribution to approximate the variational posterior $q_{\boldsymbol{\phi}}(\boldsymbol{\lambda}|\boldsymbol{z}, \boldsymbol{x})$. Accordingly, we get the ELBO2 for Hierarchical Variational Inference as

$$\text{ELBO2}(\boldsymbol{\theta}, \boldsymbol{\phi}; \boldsymbol{x}) = \mathbb{E}_{q_{\boldsymbol{\phi}}(\boldsymbol{z}, \boldsymbol{\lambda}|\boldsymbol{x})}\Big[\log p_{\boldsymbol{\theta}}(\boldsymbol{x}, \boldsymbol{z}) - \log q_{\boldsymbol{\phi}}(\boldsymbol{z}, \boldsymbol{\lambda}|\boldsymbol{x}) + \log r_{\boldsymbol{\omega}}(\boldsymbol{\lambda}|\boldsymbol{z}, \boldsymbol{x})\Big].$$

Note similar to (31), the unnecessary "0" term related to $\log q_{\boldsymbol{\phi}}(\boldsymbol{z}, \boldsymbol{\lambda}|\boldsymbol{x})$ should also be removed. Accordingly, we have

$$\nabla_{\boldsymbol{\phi}}\text{ELBO2}(\boldsymbol{\theta}, \boldsymbol{\phi}; \boldsymbol{x}) = \int \big[\nabla_{\boldsymbol{\phi}} q_{\boldsymbol{\phi}}(\boldsymbol{z}, \boldsymbol{\lambda}|\boldsymbol{x})\big]\big[\log p_{\boldsymbol{\theta}}(\boldsymbol{x}, \boldsymbol{z}) - \log q_{\boldsymbol{\phi}}(\boldsymbol{z}, \boldsymbol{\lambda}|\boldsymbol{x}) + \log r_{\boldsymbol{\omega}}(\boldsymbol{\lambda}|\boldsymbol{z}, \boldsymbol{x})\big]d\boldsymbol{\lambda}dz.$$

Obviously, Theorem 3 is readily applicable to provide GO gradients.

## H    DETAILS OF GAMMA AND NB ONE-DIMENSIONAL SIMPLE EXAMPLES

We first consider illustrative one-dimensional "toy" problems, to examine the GO gradient in Theorem 1 for both continuous and discrete random variables.

The optimization objective is expressed as

$$\max_{\boldsymbol{\phi}} \text{ELBO}(\boldsymbol{\phi}) = \mathbb{E}_{q_{\boldsymbol{\phi}}(z)}[\log p(z|x) - \log q_{\boldsymbol{\phi}}(z)] + \log p(x),$$

where for continuous $z$ we assume $p(z|x) = \text{Gam}(z; \alpha_0, \beta_0)$ for set $(\alpha_0, \beta_0)$, with $q_{\boldsymbol{\phi}}(z) = \text{Gam}(z; \alpha, \beta)$ and $\boldsymbol{\phi} = \{\alpha, \beta\}$; for discrete $z$ we assume $p(z|x) = \text{NB}(z; r_0, p_0)$ for set $(r_0, p_0)$, with $q_{\boldsymbol{\phi}}(z) = \text{NB}(z; r, p)$ and $\boldsymbol{\phi} = \{r, p\}$. Stochastic gradient ascent with one-sample-estimated gradients is used to optimize the objective, which is equivalent to minimizing $\text{KL}(q_{\boldsymbol{\phi}}(z)\|p(z|x))$.

Figure 8 shows the experimental results. For the nonnegative continuous $z$ associated with the gamma distribution, we compare our GO gradient with GRep (Ruiz et al., 2016), RSVI (Naesseth et al., 2016), and their modified version using the "sticking" idea (Roeder et al., 2017), denoted as GRep-Stick and RSVI-Stick respectively. For RSVI and RSVI-Stick, the shape augmentation parameter is set as 5 by default. The only difference between GRep and GRep-Stick (also RSVI and RSVI-Stick) is the latter does NOT analytically express the entropy $\mathbb{E}_{q_{\boldsymbol{\phi}}(z)}[-\log q_{\boldsymbol{\phi}}(z)]$. One should apply sticking because $(i)$ Figures 8(a)-8(c) clearly show its utility in reducing variance; and $(ii)$ without it, GRep and RSVI exhibit high variance that unstabilizes the optimization for small gamma shape parameters, as shown in Figures 8(d)-8(f). We adopt the sticking approach hereafter for all the compared methods. Since the gamma rate parameter $\beta$ is reparameterizable, its gradient calculation is the same for all sticking methods, including GO, GRep-Stick, and RSVI-Stick. Therefore, similar variances are observed in Figures 8(b) and 8(e). Among methods with sticking, GO exhibits the lowest variance in general, as shown in Figures 8(a) and 8(d). Note it is high variance that causes optimization issues. As a result, GO empirically provides more stable learning curves, as shown in Figures 8(c) and 8(f). For the discrete case corresponding to the NB distribution, GO is compared to REINFORCE (Williams, 1992). To estimate gradient, REINFORCE uses 1 sample of $z$ and 1 evaluation of the expected function; whereas GO uses 1 sample and 2 evaluations. To address the concern about comparison with the same number of evaluations of the expected function, another curve of REINFORCE using 2 samples (thus 2 evaluations of the expected function) is also added, termed REINFORCE2. It is apparent from Figures 8(g)-8(l) that, thanks to analytically removing the "0" terms, the GO gradient has much lower variance and thus faster convergence, even in this simple one-dimensional case.

## I    DETAILS OF THE DISCRETE VAE EXPERIMENT

Complementing the discrete VAE experiment of the main manuscript, we present below its experimental settings, implementary details, and additional results.

Since the presented statistical back-propagation in Theorem 3 of the main manuscript cannot handle discrete internal variables, we focus the single-latent-layer settings (1 layer of 200 Bernoulli random variables) for fairness, *i.e.,*

$$p_{\boldsymbol{\theta}}(\boldsymbol{x}, \boldsymbol{z}) : \boldsymbol{x} \sim \text{Bern}\big(\text{NN}_{\boldsymbol{P}_{\boldsymbol{x}|\boldsymbol{z}}}(\boldsymbol{z})\big), \boldsymbol{z} \sim \text{Bern}\big(\boldsymbol{P}_{\boldsymbol{z}}\big)$$
$$q_{\boldsymbol{\phi}}(\boldsymbol{z}|\boldsymbol{x}) : \boldsymbol{z} \sim \text{Bern}\big(\text{NN}_{\boldsymbol{P}_{\boldsymbol{z}|\boldsymbol{x}}}(\boldsymbol{x})\big).$$

(32)

Referring to the experimental settings in Grathwohl et al. (2017), we consider

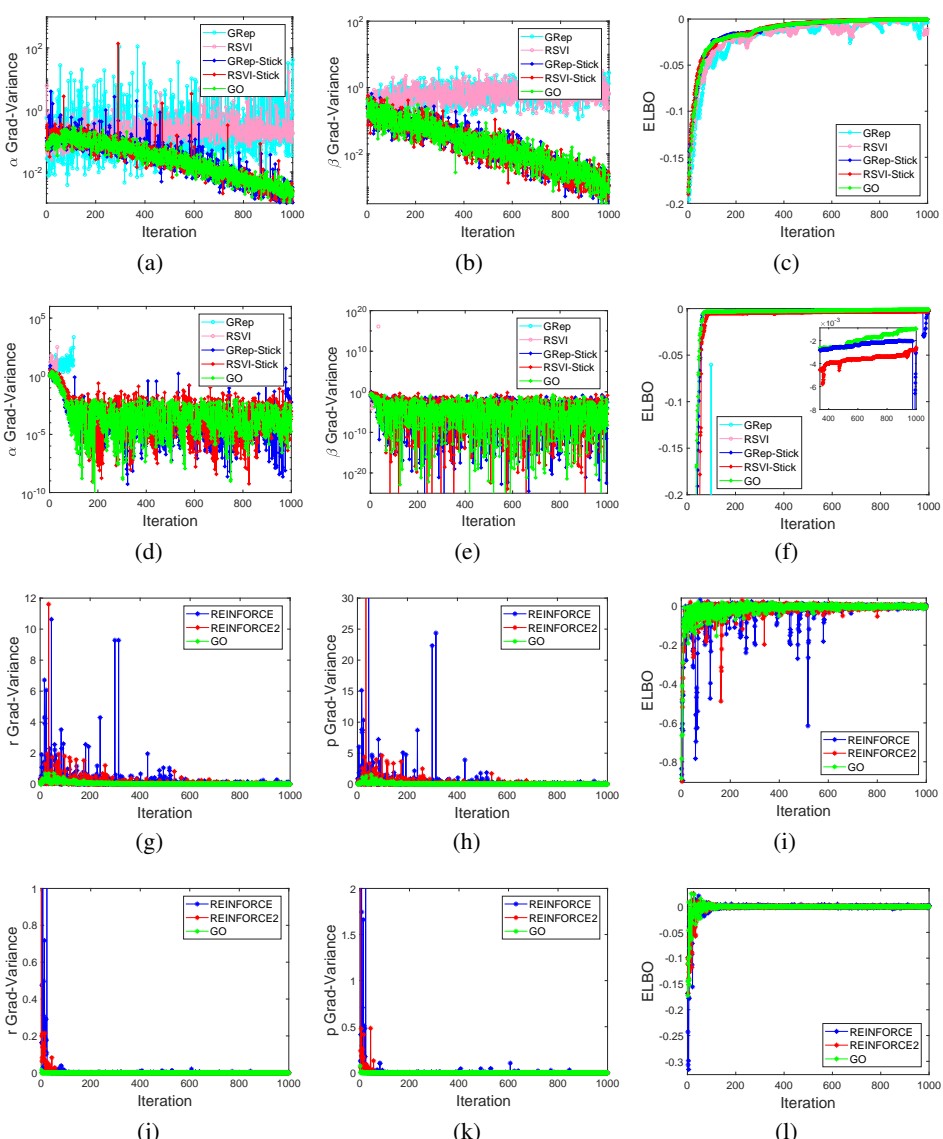

Figure 8: Gamma (a-f) and NB (g-l) toy experimental results. Columns show the gradient variance for the first parameter (gamma $\alpha$ or NB $r$), that for the second parameter (gamma $\beta$ or NB $p$), and the ELBO, respectively. The first two rows correspond to the gamma toys with posterior parameters $\alpha_0 = 1, \beta_0 = 0.5$ and $\alpha_0 = 0.01, \beta_0 = 0.5$, respectively. The last two rows show NB toy results with $r_0 = 10, p_0 = 0.2$ and $r_0 = 0.5, p_0 = 0.2$, respectively. In each iteration, gradient variances are estimated with 20 Monte Carlo samples (each sample corresponds to one gradient estimate), among which the last one is used to update parameters. 100 Monte Carlo samples are used to calculate the ELBO in the NB toys.

- **1-layer linear model:**

$$\text{NN}_{\boldsymbol{P}_{\boldsymbol{x}|\boldsymbol{z}}}(\boldsymbol{z}) = \sigma(\mathbf{W}_p^T \boldsymbol{z} + \boldsymbol{b}_p)$$
$$\text{NN}_{\boldsymbol{P}_{\boldsymbol{z}|\boldsymbol{x}}}(\boldsymbol{x}) = \sigma(\mathbf{W}_q^T \boldsymbol{x} + \boldsymbol{b}_q)$$

where $\sigma(\cdot)$ is the sigmoid function.

- **Nonlinear model:**

$$\text{NN}_{\boldsymbol{P}_{\boldsymbol{x}|\boldsymbol{z}}}(\boldsymbol{z}) = \sigma(\mathbf{W}_{p2}^T \boldsymbol{h}_p^{(2)} + \boldsymbol{b}_{p2}), \boldsymbol{h}_p^{(2)} = \tanh(\mathbf{W}_{p1}^T \boldsymbol{h}_p^{(1)} + \boldsymbol{b}_{p1}), \boldsymbol{h}_p^{(1)} = \tanh(\mathbf{W}_p^T \boldsymbol{z} + \boldsymbol{b}_p)$$
$$\text{NN}_{\boldsymbol{P}_{\boldsymbol{z}|\boldsymbol{x}}}(\boldsymbol{x}) = \sigma(\mathbf{W}_{q2}^T \boldsymbol{h}_q^{(2)} + \boldsymbol{b}_{q2}), \boldsymbol{h}_q^{(2)} = \tanh(\mathbf{W}_{q1}^T \boldsymbol{h}_q^{(1)} + \boldsymbol{b}_{q1}), \boldsymbol{h}_q^{(1)} = \tanh(\mathbf{W}_q^T \boldsymbol{x} + \boldsymbol{b}_q)$$

where $\tanh(\cdot)$ is the hyperbolic-tangent function.

The used datasets and other experimental settings, including the hyperparameter search strategy, are the same as those in Grathwohl et al. (2017).

For such single-latent-layer settings, it is obvious that Theorem 3 (also Theorem 1) can be straightforwardly applied. However, since Bernoulli distribution has finite support, as mentioned in the main manuscript, we should analytically express some expectations for lower variance, as detailed below. Notations of (8) and (9) of the main manuscript are used for clarity and also for generalization.

In fact, we should take a step back and start from (8) of the main manuscript, which is equivalent to analytically express an expectation in (9), namely

$$\nabla_{\boldsymbol{\gamma}}\mathbb{E}_{q_{\boldsymbol{\gamma}}(\boldsymbol{y})}[f(\boldsymbol{y})] = \sum_v \mathbb{E}_{q_{\boldsymbol{\gamma}}(\boldsymbol{y}_{-v})}\Big[ -\sum_{y_v}[\nabla_{\boldsymbol{\gamma}}Q_{\boldsymbol{\gamma}}(y_v)][f(\boldsymbol{y}_{-v}, y_v + 1) - f(\boldsymbol{y})]\Big], \quad (33)$$

where $Q_{\boldsymbol{\gamma}}(y_v) = \begin{cases} 1 - P_v(\boldsymbol{\gamma}) & y_v = 0 \\ 1 & y_v = 1 \end{cases}$ with $P_v(\boldsymbol{\gamma})$ being the Bernoulli probability of Bernoulli random variable $y_v$. Accordingly, we have

$$\begin{aligned}
\nabla_{\boldsymbol{\gamma}}\mathbb{E}_{q_{\boldsymbol{\gamma}}(\boldsymbol{y})}[f(\boldsymbol{y})] &= -\sum_v \mathbb{E}_{q_{\boldsymbol{\gamma}}(\boldsymbol{y}_{-v})}\Big[[\nabla_{\boldsymbol{\gamma}}Q_{\boldsymbol{\gamma}}(y_v = 0)][f(\boldsymbol{y}_{-v}, y_v = 1) - f(\boldsymbol{y}_{-v}, y_v = 0)]\Big] \\
&= \sum_v \mathbb{E}_{q_{\boldsymbol{\gamma}}(\boldsymbol{y}_{-v})}\Big[[\nabla_{\boldsymbol{\gamma}}P_v(\boldsymbol{\gamma})][f(\boldsymbol{y}_{-v}, y_v = 1) - f(\boldsymbol{y}_{-v}, y_v = 0)]\Big] \\
&= \mathbb{E}_{q_{\boldsymbol{\gamma}}(\boldsymbol{y})}\Big[\sum_v [\nabla_{\boldsymbol{\gamma}}P_v(\boldsymbol{\gamma})][f(\boldsymbol{y}_{-v}, y_v = 1) - f(\boldsymbol{y}_{-v}, y_v = 0)]\Big]
\end{aligned}$$

$$(34)$$

For better understanding only, with abused notations $\nabla_{\boldsymbol{\gamma}}\boldsymbol{P} = [\cdots, \nabla_{\boldsymbol{\gamma}}P_v(\boldsymbol{\gamma}), \cdots]^T$, $\nabla_{\boldsymbol{P}}\boldsymbol{y} = \mathbf{I}$, and $\nabla_{\boldsymbol{y}}f(\boldsymbol{y}) = [\cdots, \{f(\boldsymbol{y}_{-v}, y_v = 1) - f(\boldsymbol{y}_{-v}, y_v = 0)\}, \cdots]^T$, one should observe a chain rule within the above equation.

To assist better understanding of how to practically cooperate the presented GO gradients with deep learning frameworks like TensorFlow or PyTorch, we take (34) as an example, and present for it the following simple algorithm.

---

**Algorithm 1** An algorithm for (34) as an example to demonstrate how to practically cooperate GO gradients with deep learning frameworks like TensorFlow or PyTorch. One sample is assumed for clarity. Practically, an easy-to-use trick for changing gradients of any function $h(\boldsymbol{x})$ is to define $\hat{h}(\boldsymbol{x}) = \boldsymbol{x}^T \text{StopGradient}[\boldsymbol{g}] + \text{StopGradient}[h(\boldsymbol{x}) - \boldsymbol{x}^T\boldsymbol{g}]$ with $\boldsymbol{g}$ the desired gradients.

---

\# Forward-Propagation
1. $\boldsymbol{\gamma} \to \boldsymbol{P}(\boldsymbol{\gamma})$: Calculate Bernoulli probabilities $\boldsymbol{P}(\boldsymbol{\gamma})$
2. $\boldsymbol{P}(\boldsymbol{\gamma}) \to \boldsymbol{y}$: Sample $\boldsymbol{y} \sim \text{Bern}(\boldsymbol{P}(\boldsymbol{\gamma}))$           ▷ Change gradient: $\nabla_{\boldsymbol{P}(\boldsymbol{\gamma})}\boldsymbol{y} = \mathbf{I}$
3. $\boldsymbol{y} \to f(\boldsymbol{y})$: Calculate the loss $f(\boldsymbol{y})$
                    ▷ Change gradient: $[\nabla_{\boldsymbol{y}}f(\boldsymbol{y})]_v = f(\boldsymbol{y}_{-v}, y_v = 1) - f(\boldsymbol{y}_{-v}, y_v = 0)$
\# Back-Propagation
1. Rely on the mature auto-differential software for back-propagating gradients

---

For efficient implementation of $\mathbb{D}_{\boldsymbol{y}}f(\boldsymbol{y})$, one should exploit the prior knowledge of function $f(\boldsymbol{y})$. For example, $f(\boldsymbol{y})$s are often neural-network-parameterized. Under that settings, one should be able to exploit tensor operation to design efficient implementation of $\mathbb{D}_{\boldsymbol{y}}f(\boldsymbol{y})$. Again we take (34) as an example, and assume $f(\boldsymbol{y})$ has the special structure

$$f(\boldsymbol{y}) = r(\boldsymbol{\Theta}^T\boldsymbol{h} + \boldsymbol{c}), \boldsymbol{h} = \sigma(\mathbf{W}^T\boldsymbol{y} + \boldsymbol{b}) \quad (35)$$

where $\sigma(\cdot)$ is an element-wise nonlinear activation function, and $r(\cdot)$ is a function that takes in a vector and outputs a scalar. One can easily modify the above $f(\boldsymbol{y})$ for the considered discrete VAE experiment.

Since $y_v$s are now Bernoulli random variables with support $\{0, 1\}$, we have that

$$\begin{aligned}
\big[\mathbb{D}_{\boldsymbol{y}}f(\boldsymbol{y})\big]_v &= f(\boldsymbol{y}_{-v}, y_v = 1) - f(\boldsymbol{y}_{-v}, y_v = 0) \\
&= \begin{cases} f(\boldsymbol{y}_{-v}, y_v + 1) - f(\boldsymbol{y}) & y_v = 0 \\ f(\boldsymbol{y}) - f(\boldsymbol{y}_{-v}, y_v - 1) & y_v = 1 \end{cases} \\
&= a_v[f(\boldsymbol{y}_{-v}, y_v + a_v) - f(\boldsymbol{y})],
\end{aligned}$$

$$(36)$$

where $a_v = \begin{cases} 1 & y_v = 0 \\ -1 & y_v = 1 \end{cases}$ is the $v$-th element of vector $\boldsymbol{a}$.

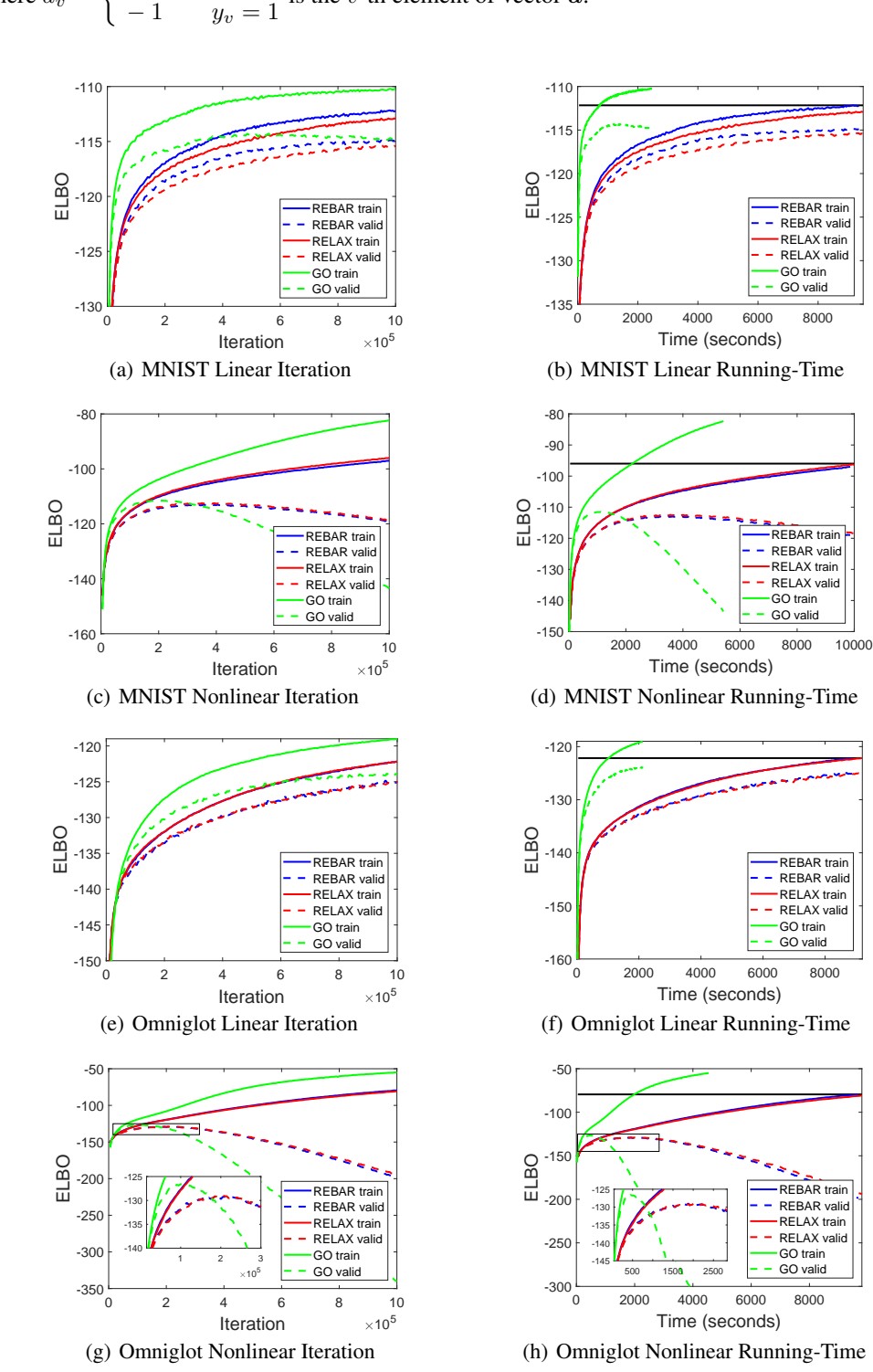

(a) MNIST Linear Iteration

(b) MNIST Linear Running-Time

(c) MNIST Nonlinear Iteration

(d) MNIST Nonlinear Running-Time

(e) Omniglot Linear Iteration

(f) Omniglot Linear Running-Time

(g) Omniglot Nonlinear Iteration

(h) Omniglot Nonlinear Running-Time

Figure 9: Training/Validation ELBOs for the discrete VAE experiments. Rows correspond to the experimental results on the MNIST/Omniglot dataset with the 1-layer-linear/nonlinear model, respectively. Shown in the first/second column is the ELBO curves as a function of iteration/running-time. All methods are run with the same learning rate for $1,000,000$ iterations. The black line represents the best training ELBO of REBAR and RELAX. ELBOs are calculated using all training/validation data. Note GO does *not* suffer more from over-fitting, as clarified in the text.

Table 4: Average running time per 100 iterations for discrete variational autoencoders. Results of REBAR and RELAX are obtained by running the released code[5] from Grathwohl et al. (2017). The same computer with one Titan Xp GPU is used.

| Dataset | Model | REBAR | RELAX | GO |
|---------|-------|-------|-------|-----|
| **MNIST** | Linear 1 layer | 0.94s | 0.95s | **0.25**s |
| | Nonlinear | 0.99s | 1.02s | **0.54**s |
| **Omniglot** | Linear 1 layer | 0.89s | 0.92s | **0.21**s |
| | Nonlinear | 0.97s | 0.98s | **0.45**s |

Then to efficiently calculate the $f(\boldsymbol{y}_{-v}, y_v + a_v)$s, we use the following batch processing procedure to benefit from parallel computing.

- Step 1: Define $\Xi_{\boldsymbol{y}}\boldsymbol{h}$ as the matrix whose element $[\Xi_{\boldsymbol{y}}\boldsymbol{h}]_{vj}$ represents the "new" $h_j^*$ when input $\{\boldsymbol{y}_{-v}, y_v + a_v\}$ in (35). Then, we have

$$[\Xi_{\boldsymbol{y}}\boldsymbol{h}]_{vj} = \sigma(\boldsymbol{y}^T\mathbf{W}_{:j} + \boldsymbol{b}_j + a_v\mathbf{W}_{vj})$$

  where $\mathbf{W}_{:j}$ is the $j$-th column of the matrix $\mathbf{W}$. Note the $v$th row of $\Xi_{\boldsymbol{y}}\boldsymbol{h}$, *i.e.*, $[\Xi_{\boldsymbol{y}}\boldsymbol{h}]_{v:}$, happens to be the "new" $\boldsymbol{h}^*$ when input $\{\boldsymbol{y}_{-v}, y_v + a_v\}$.

- Step 2: Similarly, we define $\Xi_{\boldsymbol{y}}f$ as the vector whose element $[\Xi_{\boldsymbol{y}}f]_v = f(\boldsymbol{y}_{-v}, y_v + a_v)$. Utilizing $\Xi_{\boldsymbol{y}}\boldsymbol{h}$ obtained in Step 1, we have

$$\Xi_{\boldsymbol{y}}f = r([\Xi_{\boldsymbol{y}}\boldsymbol{h}]\boldsymbol{\vartheta} + \boldsymbol{c}^T),$$

  where $r(\cdot)$ is applied to each row of the matrix $([\Xi_{\boldsymbol{y}}\boldsymbol{h}]\boldsymbol{\vartheta} + \boldsymbol{c}^T)$.

Note the above batch processing procedure can be easily extended to deeper neural networks. Accordingly, we have

$$\mathbb{D}_{\boldsymbol{y}}f(\boldsymbol{y}) = \boldsymbol{a} \odot [\Xi_{\boldsymbol{y}}f - f(\boldsymbol{y})],$$

where $\odot$ represents the matrix element-wise product.

Now we can rely on Algorithm 1 to solve the problem whose objective has its gradient expressed as (34), for example the inference of the single-latent-layer discrete VAE in (32).

All training curves versus iteration/running-time are given in Figure 9, where it is apparent that GO provides better performance, a faster convergence rate, and a better running efficiency in all situations. The average running time per 100 iterations for the compared methods are given in Table 4, where GO is $2 - 4$ times faster in finishing the same number of training iterations. We also quantify the running efficiency of GO by considering its running time to achieve the best training ELBO (within $1,000,000$ training iterations) of RERAR/RELAX, referring to the black lines shown in the second-column subfigures of Figure 9. It is clear that GO is approximately $5 - 10$ times more efficient than REBAR/RELAX in the considered experiments.

As shown in the second and fourth rows of Figures 9, for the experiments with nonlinear models all methods suffer from over-fitting, which originates from the redundant complexity of the adopted neural networks and appeals for model regularizations. We detailedly clarify these experimental results as follows.

- All the compared methods are given the same and only objective, namely to maximize the training ELBO on the same training dataset with the same model; GO clearly shows its power in achieving a better objective.

- The "level" of over-fitting is ultimately determined by the used dataset, model, and objective; it is independent of the adopted optimization method. Different optimization methods just reveal different optimizing trajectories, which show different sequences of training objectives and over-fitting levels (validation objectives).

---

[5]github.com/duvenaud/relax

- Since all methods are given the same dataset, model, and objective, they have the same over-fitting level. Because GO has a lower variance, and thus more powerful optimization capacity, it gets to the similar situations much faster than REBAR/RELAX. Note this does *not* mean GO suffers more from over-fitting. In fact, GO provides better validation ELBOs in all situations, as shown in Figure 9 and also Table 1 of the main manuscript. In practice, GO can benefit from the early-stopping trick to get a better generalization ability.

## J    DETAILS OF THE MULTINOMIAL GAN

Complementing the multinomial GAN experiment in the main manuscript, we present more details as follows. For a quantitative assessment of the computational complexity, our PyTorch code takes about 30 minutes to get the most challenging 4-bit task in Figs. 4 and 11, with a Titan Xp GPU.

Firstly, recall that the generator $p_{\boldsymbol{\theta}}(\boldsymbol{x})$ of the developed multinomial GAN (denoted as MNGAN-GO) is expressed as

$$\boldsymbol{\epsilon} \sim \mathcal{N}(\mathbf{0}, \mathbf{I}), \boldsymbol{x} \sim \mathrm{Mult}(\mathbf{1}, \mathrm{NN}_{\mathbf{P}}(\boldsymbol{\epsilon})),$$

where $\mathrm{NN}_{\mathbf{P}}(\boldsymbol{\epsilon})$ denotes use of a neural network to project $\boldsymbol{\epsilon}$ to distribution parameters $\mathbf{P}$. For brevity, we integration the generator's parameters $\boldsymbol{\theta}$ into the NN notation and do not explicitly express them. Multinomial leaf variables $\boldsymbol{x}$ is used to describe discrete observations with a finite alphabet. To train MNGAN-GO, the vanilla GAN loss (Goodfellow et al., 2014) is used. A deconvolutional neural network as in Radford et al. (2015) is used to map $\boldsymbol{\epsilon}$ to $\mathbf{P}$ in the generator. The discriminator is constructed as a multilayer perceptron. Detailed model architectures are given in Table 5. Figure 10 illustrates the pipeline of MNGAN-GO. Note MNGAN-GO has a smaller number of parameters, compared to BGAN (Hjelm et al., 2018).

For clarity, we briefly discuss the employed data preprocessing. Taking MNIST for an example, the original data are 8-bit grayscale images, with pixel intensities ranging from $0$ to $255$. For the $n$-bit experiment, we obtain the real data, such as the 2-bit one in Figure 10, by rescaling and quantizing the pixel intensities to the range $[0, 2^n - 1]$, having $2^n$ different states (values).

For the 1-bit special case, the multinomial distribution reduces to the Bernoulli distribution. Of course, one could intuitively employ the redundant multinomial distribution, which is denoted as 1-bit (2-state) in Table 2. An alternative and popular approach is to adopt the Bernoulli distribution to remove the redundancy by only modeling its probability parameters; we denote this case as 1-bit (1-state, Bernoulli) in Table 2.

Figure 11 shows the generated samples from the compared models on different quantized MNIST. It is obvious that MNGAN-GO provides images with better quality and wider diversity in general.

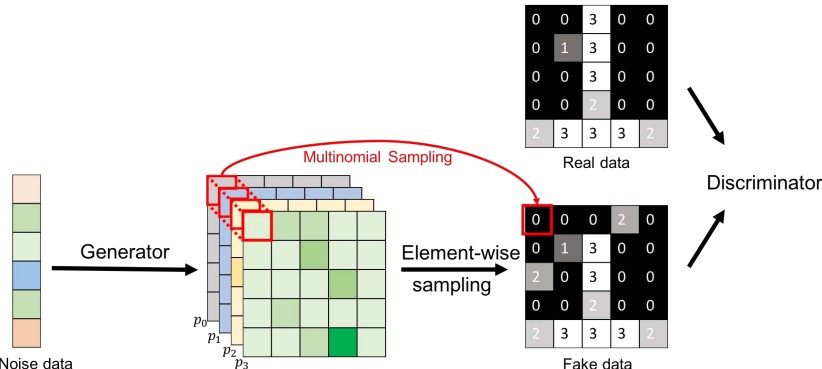

Figure 10: Illustration of MNGAN-GO.

## K    DETAILS OF HVI FOR DEF AND DLDA

Complementing the HVI experiments in the main manuscript, we present more details as follows.

Table 5: Model Architectures of the Multinomial GAN.

| Generator | Discriminator |
|---|---|
| Gaussian noise (100 dimension) | Input Image |
| $4 \times 4$ conv. 256 lReLU, stride 1, zero-pad 0, BN | Linear output 512, lReLU, SN |
| $4 \times 4$ conv. 128 lReLU, stride 2, zero-pad 1, BN | Linear output 256, lReLU, SN |
| $4 \times 4$ conv. 64 lReLU, stride 2, zero-pad 2, BN | Linear output 128, lReLU, SN |
| $4 \times 4$ conv. $2^{bit}$ SoftMax, stride 2, zero-pad 1, BN | Linear output 1, Sigmoid |
| Multinomial Sampling. Output: $28 \times 28$ | |

Table 6: Inception scores on quantized MNIST. BGAN's results are ran with the author released code `https://github.com/rdevon/BGAN`.

| bits (states) | BGAN | MNGAN-GO |
|---|---|---|
| 1 (1, Bernoulli) | $8.31 \pm .06$ | $\mathbf{9.10 \pm .06}$ |
| 1 (2) | $\mathbf{8.56 \pm .04}$ | $8.40 \pm .07$ |
| 2 (4) | $7.76 \pm .04$ | $\mathbf{9.02 \pm .06}$ |
| 3 (8) | $7.26 \pm .03$ | $\mathbf{9.26 \pm .07}$ |
| 4 (16) | $6.29 \pm .05$ | $\mathbf{9.27 \pm .06}$ |

For a quantitative assessment of the computational complexity, the TensorFlow code used takes about $0.08$ seconds per iteration (including $> 40,000$ Meijer-G function calculations, with an approximate algorithm coded also with TensorFlow).

Deep exponential families (DEF) (Ranganath et al., 2015) is expressed as

$$\boldsymbol{x} \sim \text{Pois}(\mathbf{W}^{(1)}\boldsymbol{z}^{(1)}), \boldsymbol{z}^{(l)} \sim \text{Gam}\big(\alpha_z, {}^{\alpha_z}/\mathbf{w}^{(l+1)}\boldsymbol{z}^{(l+1)}\big), \mathbf{W}^{(l)} \sim \text{Gam}(\alpha_0, \beta_0),$$

where $\alpha_z = 0.1$, $\alpha_0 = 0.3$, and $\beta_0 = 0.1$ following Ruiz et al. (2016); Naesseth et al. (2016). Since the variables of interest, $\mathbf{W}^{(l)}$ and $\boldsymbol{z}^{(l)}$, are gamma-distributed, we design the variational approximations as

$$q_{\phi_z}(\boldsymbol{z}|\boldsymbol{x}) : \boldsymbol{z}^{(1)} \sim \text{Gam}\big(\text{NN}_{\boldsymbol{\alpha}}^{(1)}(\boldsymbol{x}), \text{NN}_{\boldsymbol{\beta}}^{(1)}(\boldsymbol{x})\big), \boldsymbol{z}^{(l)} \sim \text{Gam}\big(\text{NN}_{\boldsymbol{\alpha}}^{(l)}(\boldsymbol{z}^{(l-1)}), \text{NN}_{\boldsymbol{\beta}}^{(l)}(\boldsymbol{z}^{(l-1)})\big),$$

$$q_{\phi_\mathbf{W}}(\mathbf{W}) : \mathbf{W}^{(l)} \sim \text{Gam}\big(\boldsymbol{\alpha}_{\mathbf{W}}^{(l)}, \boldsymbol{\beta}_{\mathbf{W}}^{(l)}\big),$$

where $\text{NN}^{(l)}(\cdot)$s are set to the same shapes as the corresponding $\boldsymbol{z}^{(l)}$, $\boldsymbol{\alpha}_{\mathbf{W}}^{(l)}$ and $\boldsymbol{\beta}_{\mathbf{W}}^{(l)}$ also have the shape of $\mathbf{W}^{(l)}$. We employ a simple two-layer DEF for demonstration, with $\boldsymbol{z}^{(1)}$ and $\boldsymbol{z}^{(2)}$ having 128 and 64 components, respectively. The mini-batch size is set to 200. One-sample gradient estimates are used to train the model for the compared methods. For RSVI (Naesseth et al., 2016), the shape augmentation parameter $B$ is set to 5. All experimental settings, except for different ways to calculate gradients, are the same for the compared methods.

Deep latent Dirichlet allocation (DLDA) (Zhou et al., 2015; 2016; Cong et al., 2017) is expressed as

$$\begin{aligned}
\boldsymbol{x} \sim &\text{Pois}(\boldsymbol{\Phi}^{(1)}\boldsymbol{z}^{(1)}), \boldsymbol{z}^{(l)} \sim \text{Gam}\big(\boldsymbol{\Phi}^{(l+1)}\boldsymbol{z}^{(l+1)}, c^{(l+1)}\big), \boldsymbol{z}^{(L)} \sim \text{Gam}\big(\boldsymbol{r}, c^{(L+1)}\big), \\
&\boldsymbol{\Phi}^{(l)} \sim \text{Dir}(\eta_0), c^{(l)} \sim \text{Gam}(e_0, f_0), \boldsymbol{r} \sim \text{Gam}(\gamma_0/K, c_0),
\end{aligned} \tag{37}$$

where hyperparameters are chosen following Cong et al. (2017). Compared to DEF, DLDA is more complicated: 1) model is constructed via the highly nonlinear Gamma shape parameters; 2) dictionaries are described by more challenging Dirichlet distributions; 3) more interested random variables.

The variational approximations for DLDA are designed as

$$q_{\boldsymbol{\phi}}(\boldsymbol{z}, c|\boldsymbol{x}) : \boldsymbol{z}^{(1)} \sim \text{Gam}\big(\text{NN}_{\boldsymbol{\alpha}_z}^{(1)}(\boldsymbol{x}), \text{NN}_{\boldsymbol{\beta}_z}^{(1)}(\boldsymbol{x})\big), c^{(2)} \sim \text{Gam}\big(\text{NN}_{\boldsymbol{\alpha}_c}^{(2)}(\boldsymbol{z}^{(1)}), \text{NN}_{\boldsymbol{\beta}_c}^{(2)}(\boldsymbol{z}^{(1)})\big),$$

$$\boldsymbol{z}^{(l)} \sim \text{Gam}\big(\text{NN}_{\boldsymbol{\alpha}_z}^{(l)}(\boldsymbol{z}^{(l-1)}), \text{NN}_{\boldsymbol{\beta}_z}^{(l)}(\boldsymbol{z}^{(l-1)})\big), c^{(l+1)} \sim \text{Gam}\big(\text{NN}_{\boldsymbol{\alpha}_c}^{(l+1)}(\boldsymbol{z}^{(l)}), \text{NN}_{\boldsymbol{\beta}_c}^{(l+1)}(\boldsymbol{z}^{(l)})\big),$$

$$q_{\boldsymbol{\phi}}(\boldsymbol{\Phi}) : \boldsymbol{\Phi}^{(l)} \sim \text{Dir}(\boldsymbol{\eta}^{(l)}),$$

$$q_{\boldsymbol{\phi}}(\boldsymbol{r}) : \boldsymbol{r} \sim \text{Gam}(\boldsymbol{\alpha}_r, \boldsymbol{\beta}_r),$$

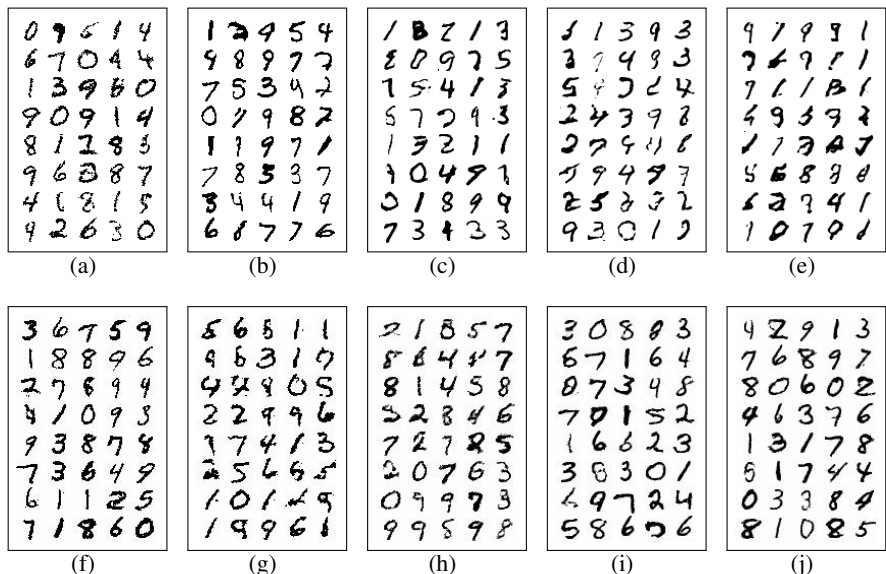

Figure 11: Generated images from BGAN (top) and MNGAN-GO (bottom). Columns correspond to 1-bit(Bernoulli), 1-bit, 2-bit, 3-bit, 4-bit tasks, respectively.

where motivated by the original upward-downward Gibbs sampler developed in Zhou et al. (2015), we specify $\text{NN}_{\beta_z}^{(l)}(\cdot)$ as a scaler to mimic the Gibbs conditional posteriors. The other $\text{NN}(\cdot)$s and also $\eta^{(l)}, \alpha_r, \beta_r$ have the same shapes of the corresponding random variables. A three-layer DLDA, having 128, 64, 32 components for latent $z^{(1)}$, $z^{(2)}$, $z^{(3)}$ respectively, is trained on MNIST with mini-batches of size 200.

With the learned variational inference nets, one could efficiently project the observed $x$ to its latent variables $z, c$ during testing. For applications requiring realtime processing, this is a clear advantage. For demonstration, Figure 12 shows test data samples $x$ and their reconstruction

$$\hat{x} = \Phi^{(1)}\hat{z}^{(1)}, \hat{z}^{(1)} \sim \text{Gam}\big(\text{NN}_{\alpha_z}^{(1)}(x), \text{NN}_{\beta_z}^{(1)}(x)\big),$$

where one Monte Carlo sample is used to calculate $\hat{z}^{(1)}$.

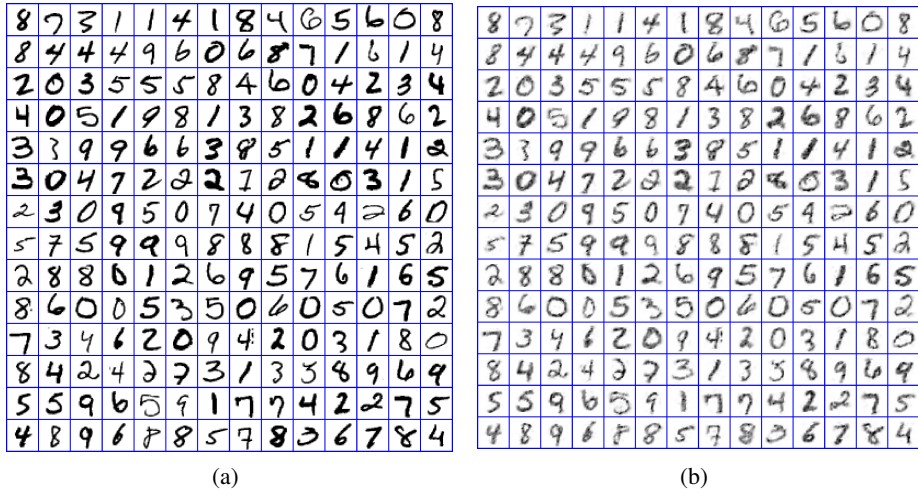

Figure 12: (a) Test data samples and (b) their reconstruction via the learned 128-64-32 DLDA.

Note for the challenging DLDA task in (37), we find it tricky to naively apply pure-gradient-based learning methods. The main reason is: the latent code $z^{(l)}$s and their gamma shape parameters $\Phi^{(l+1)}z^{(l+1)}$s are usually extremely sparse, meaning most elements are almost zero; a gamma distribution $z \sim \mathrm{Gam}(\alpha, \beta)$ with almost-zero $\alpha$ has an increasingly steep slope when $z$ approaches zero, namely the gradient wrt $z$ shall have an enormous magnitude that unstablize the learning procedure. Even though it might not be sufficient to just use the first-order gradient information, empirically the following tricks help us get the presented reasonable results.

- Let $z^{(l)} \geq T_z$, where $T_z = 1e^{-5}$ is used in the experiments;
- Let $c^{(l)} \geq T_c$, where $T_c = 1e^{-5}$;
- Let $\Phi^{(l+1)}z^{(l+1)} \geq T_\alpha$ with $T_\alpha = 0.2$;
- Use a factor to compromise the likelihood and prior for each $z^{(l)}$.

For more details, please refer to our released code. We are working on exploiting higher-order information (such as Hessian) to help remedy this issue.

