# OpenReview forum: "GO Gradient for Expectation-Based Objectives"
_ICLR.cc/2019/Conference_

### Official Review · AnonReviewer3 · 2018-11-02
**Reasonable methods but some unclear points**

**Rating:** 6
**Confidence:** 4

**Review:**

The paper design a low variance gradient for distributions associated with continuous or discrete random variables. The gradient is designed in the way to approximate the  property of reparameterization gradient.  The paper is comprehensive and includes mathematical details.

I have following comments/questions

1. What is the \kappa in “variable-nabla” stands for? What is the gradient w.r.t. \kappa?

2. In Eq(8), does the outer expectation w.r.t . y_{-v} be approximated by one sample? If so, it is using the local expectation method. How does that differs from Titsias & Lazaro-Gredilla(2015) both mathematically and experimentally?

3. Assume y_v is M-way categorical distribution, Eq(8) evaluates f by 2*V*M times which can be computationally expensive. What is the computation complexity of GO? How to explain the fast speed shown in the experiments?

4. A most simple way to reduce the variance of REINFORCE gradient is to take multiple Monte-Carlo samples at the cost of more computation with multiple function f evaluations. Assume GO gradient needs to evaluate f N times, how does the performance compared with the REINFORCE gradient with N Monte-Carlo samples?

5. In the discrete VAE experiment, upon brief checking the results in Grathwohl(2017), it shows validation ELBO for MNIST as (114.32,111.12), OMNIGLOT as (122.11,128.20) from which two cases are better than GO. Does the hyper parameter setting favor the GO gradient in the reported experiments? Error bar may also be needed for comparison. What about the performance of GO gradient in the 2 stochastic layer setting in Grathwohl(2017)?

6. The paper claims GO has less parameters than REBAR/RELAX. But in Figure 9, GO has more severe overfitting. How to explain this contradicts between the model complexity and overfitting?

---

> ### Author Response · Authors · 2018-11-14
> **Addressing Reviewer 3 concerns**
>
> Thank you for your time and effort of reviewing our paper. Please see our response below.
>
> \kappa is an assistant notation to remove the ambiguity of the two \gammas in G_{\gamma}^{q_{\gamma} (y)}. \kappa stands for the parameter/variable of which the gradient information is needed. For example,
> (i) g_{\kappa}^{q_{\gamma}(y)} = frac{-1}{q_{\gamma}(y)} \nabla_{\kappa} Q_{\gamma}(y)}, where \kappa is \gamma, as in Theorem 1;
> (ii) g_{\kappa}^{q_{\gamma}(y|\lambda)} = frac{-1}{q_{\gamma}(y|\lambda)} \nabla_{\kappa} Q_{\gamma}(y |\lambda), where \kappa could be \gamma or \lambda.
>
> Eqs. (7) and (8) are the foundations GO is built on, but they are not our GO. GO is defined in Eq. (9) of Theorem 1.
> For Eq. (9), yes, y_{-v} is selected from one sample y in the experiments. But GO is not the local expectation gradient (Titsias & Lazaro-Gredilla, 2015), because GO uses different information (the derivative of the CDF and the difference of the expected function). As pointed out in the last paragraph of Sec. 3, when y_v has finite support and the computational cost is acceptable, one could use the local idea from Titsias & Lazaro-Gredilla(2015) for lower variance, namely analytically evaluate a part of expectations in Eq. (9). For a detailed example, please refer to Appendix I. The main difference between the local expectation gradient and the proposed GO is that the latter is applicable to where the former might not be applicable, such as where y_v has infinite support or the computational cost for the local expectation is prohibitive.
>
> Please note our GO is defined in Eq. (9). As pointed out in the last paragraph of Sec. 3, calculating Dy[f(y)] (requiring V+1 f evaluations) could be computationally expensive. We also stated there, “for f(y) often used in practice special properties hold that can be exploited for efﬁcient parallel computing”. We took the VAE experiment in Sec 7.2 as an example and gave in Appendix I its detailed analysis/implementation, in which you might be interested. More specifically, the two bullets after Table 4, should be able to address your question on fast speed. Also, as noted in the penultimate paragraph of Sec. 7.2, less parameters (without neural-network-parameterized control variant) could be another reason for GO’s efficiency.
>
> As for computation complexity, since different random variables (RVs) have different variable-nabla (as shown in Table 3 in Appendix), GO has different computation complexity for different RVs. After choosing a specific RV, one should be able to obtain GO’s computation complexity straightforwardly. For quantitative evaluation, the running time for each experiment has been given in the corresponding Appendix. Please check there if interested.
>
> Thank you for pointing out the concern on multi-sample-based REINFORCE. We have added another curve labeled REINFORCE2 to the one-dimensional NB experiments (see Fig. 8 for complete results), where the number 2 means using 2 samples to estimate the REINFORCE gradient. In this case, REINFORCE2 uses 2 samples and 2 f evaluations in each iteration, whereas GO uses 1 sample and 2 f evaluations. As expected, REINFORCE2 still exhibits higher variance than GO even in this simple one-dimensional setting. Multi-sample-based REINFORCE for other experiments is believed unnecessary, because (i) the variance of REINFORCE is well-known to increase with dimensionality; (ii) after all, if multi-sample-based REINFORCE works well in practice, why we need variance-reduction techniques?
>
> Please refer to Sec. 7.2 and Appendix I, the author released code from Grathwohl(2017) (github.com/duvenaud/relax) were run to obtain the results of REBAR and RELAX. We adopted the same hyperparameter settings therein for our GO. So, we do not think the hyperparameter settings favor our GO in the reported experiments.
> Please refer to the first paragraph of Sec. 7.2, “Since the statistical back-propagation in Theorem 3 cannot handle discrete internal variables, we focus on the single-latent-layer settings (1 layer of 200 Bernoulli random variables).”
> If you are interested, as stated in the last paragraph of Sec 7.2, we presented in Appendix B.4 a procedure to assist our methods in handling discrete internal RVs. We believe that procedure might be useful for the inference of models with discrete internal RVs (like the multi-layer discrete VAE).
>
> Please refer to the last paragraph of Appendix I, where we explained this misunderstanding in detail. In short, GO does not suffer more from overfitting; one reason is GO can provide higher validation ELBO. Actually, we believe it is GO’s efficiency that causes this misunderstanding.
>
> We hope your concerns have been addressed. If not, further discussion would be welcomed.

---

### Official Review · AnonReviewer1 · 2018-11-02
**A solid contribution with some presentation issues: scope of applicability, clarity, technical correctness**

**Rating:** 7
**Confidence:** 4

**Review:**

* Summary

The paper proposes an improved method for computing derivatives of the expectation. Such problems arises with many probabilistic models with noises or latent variables. The paper proposes a new gradient estimator of low variance applicable in certain scenarios, in particular it allows training of generative models in which observations and/or latent variables are discrete.
The submission clearly improves the state-of-the-art, experimentally demonstrates the method on several problems comparing with the alternative techniques. In what concerns the optimization, the method achieves a better objective value much faster, confirming that it is a lower variance gradient estimator.
The clarity of the presentation (in particular the description of when the method is applicable) and the technical correctness of the paper are somewhat lacking. In terms of applicability, it seems that many cases where discrete latent variables would be really interesting are not covered (e.g. sigmoid belief networks); the paper demonstrates experiments with discrete images (binary or 4-bit) not particularly motivated in my opinion. It also contains lots of additional technical details and experiments in the appendix, which I unfortunately did not review.

* Clarity

In the abstract the paper promises more than it delivers. Many problems can be cast as optimizing an expectation-based objective. The result does not at all apply to all of them. The reparameterization trick does not apply to all continuous random variables, only to such that the reparameterization satisfies certain smoothness conditions. Discrete variables are supported by the method only in the case that the distribution factors over all discrete variables conditionally on any additional “continuous variables” (to which the reparameterization trick is applicable). This very much limits the utility of the method. In particular it is not applicable to learning e.g. sigmoid belief networks [Neal, 92] (with conditional Bernoulli units) and many other problems.

“reparametrizable distributions”
A Bernoulli(p) random variable is discrete, yet it is reparametrizable as [Z>p] with Z following standard logistic distribution, whose density and cdf is smooth.

Because of the above many discussions about discrete vs. continuous variables are missleading.

Section 2. The notation of the true distribution as “q” the model as p and the approximate posterior of the model as “q” again is inconsistent. I find the background on ELBO and GANs unnecessary occluding the clarity at this point. For the purpose of introduction, it might be better to give examples of expectation objectives such as:
- dropout: q is the distribution of NN outputs given the input image and integrating out latent dropout noises, gamma are parameters of this NN.
- VAE, GAN: q is the generative model defined as a mapping of a standard multivariate normal distribution by a NN.
- sigmoid belief networks: q is a Bayesian network where each conditional distribution is a logistic regression model.
Then to state to which of these cases the results of the paper are applicable, allow for an improvement of the variance and at what additional computational cost (considering the cost of evaluating the discrete derivatives).

Section 3.
Contrary to the discussion, there are examples of non-negative distributions to which the reparameterization trick can be applied, including log-Normal and Gamma distributions.

Method:
In the case when Rep trick is applicable, is it identical to GO? The difference seems to be only in that the mapping tau may be different from Q^-1. However, this only affects the method of drawing the samples from a fixed known distribution and should have no more effect on the results than say a choice of a pseudo-random number generator. Yet, in Fig.1 some difference is observed between the methods, why is that so?

Sec 7.1
“We adopt the sticking approach hereafter”. Does it mean it is applied with all experiments with GO?

* Related Work

The state of the art allows combining differentiable and non-differentiable pieces of computation:
[Schulman, J., Heess, N., Weber, T., Abbeel, P.: Gradient estimation using stochastic computation graphs.]
I believe it should be discussed in related work. Limitations / where the proposed method brings an improvement should be highlighted.

* Technical Correctness
Equations (5) and (6) require a theorem of differentiating under integral (expectation), such as Leibnitz rule, which in case of (6) requires q_gamma(y)f(y) to be continuous in y and q_gamma(y) continuously differentiable in gamma.
Equation (7) (integration by parts) holds only with some additional requires on f.
Theorem 1 does not take account for the above conditions.

---

> ### Author Response · Authors · 2018-11-14
> **Addressing Reviewer 1 concerns**
>
> We appreciate your time and effort of reviewing our paper, and thank you for the insightful and constructive comments.
>
> For simplicity of the main paper, we moved all the detailed proofs to the Appendix. More specifically, the proofs for Theorem 1, Lemma 1, Theorem 2, Corollary 1, and Theorem 3 are given in Appendix A, C, D, E, and F, respectively.
>
> Thanks a lot for pointing out the smoothness conditions for reparameterization; we have carefully revised our paper to remove the misleading statements and to make it clearer when our method (and also the reparameterization trick, Rep) is applicable. For your comments wrt discrete random variables (RVs), unfortunately, we haven’t found a principled way to back-propagate gradient through discrete internal RVs (like in multi-layer sigmoid belief networks). However, as stated in the last paragraph of Sec. 7.2, we presented in Appendix B.4 a procedure to assist our methods in handling discrete internal RVs. We believe that procedure could be useful for the inference of models like the multi-layer sigmoid belief networks. As for the conditional independency, it is actually removed after marginalizing out additional continuous RVs (which could be non-reparameterizable RVs like Gamma). Also note that one can strengthen the aforementioned procedure by inserting more additional continuous internal RVs into the inference model to enlarge its (marginal) description power.
>
> The notations are chosen for harmony and also to keep consistency with the main literature. For example, one can add another expectation wrt the true data distribution q(x) to the ELBO in Eq. (1), that is, E_{q(x)} [ELBO] = E_{q(x) q(z|x)} [log p(x,z) – log q(z|x)]  \propto  - KL[q(x)q(z|x) || p(x,z)].
>
> For dropout, since the dropout rate is a tunable hyperparameter that need not be learned (thus no back-propagation is required), one can use Rep to construct the q distribution you defined. If we understand correctly, in that case we cannot demonstrate our advantages. Currently, the proposed method cannot be directly applied to multi-layer sigmoid belief networks (without the procedure in Appendix B.4). We have made an explicit statement of this in the revised manuscript.
>
> Thank you for pointing this out. However, it’s believed that Rep cannot be applied to Gamma distributions [1,2]. We have revised our statement to “There are situations for which Rep is not readily applicable, e.g., where the components of y may be discrete or nonnegative Gamma distributed”.
> [1] F. Ruiz, M. Titsias, and D. Blei. The generalized reparameterization gradient. In NIPS, pp. 460–468, 2016.
> [2] C. Naesseth, F. Ruiz, S. Linderman, and D. Blei. Rejection sampling variational inference. arXiv:1610.05683, 2016.
>
> Yes, Lemma 1 shows that our deep GO will reduce to Rep when Rep is applicable. We are not sure whether you were asking about the difference in Fig. 1 or Fig. 2. So, two responses are given below.
> (A) In Fig. 1, the difference comes from the definition of node y^(i). For deterministic deep neural networks, node y^(i) is the activated value after an activation function, where deterministic chain rule can be readily applied; while for deep GO gradient, node y^(i) might be the sample of a non-reparameterizable RV, where deterministic chain rule is not applicable. Please also refer to the main contribution (ii) of our response to Reviewer 2.
> (B) If you were interested in the difference in Fig. 2 (a)(b), the reasons include (1) the standard Rep cannot be applied to Gamma RVs; (2) both GRep and RSVI are designed to approximately reparametrize Gamma RVs; (3) GO generalizes Rep to non-reparameterizable RVs; or in other words, GO is identical to the exact Rep for Gamma RVs.
>
> Yes, the sticking approach was implicitly adopted for all the compared methods when it is applicable. We have made a clear statement in the revised paper.
>
> Since stochastic computation graph (SCG) is based on REINFORCE and our method is based on GO, the comparison between SCG and our method is (roughly speaking) identical to that between REINFORCE and GO. That is, SCG is more generally applicable but with higher variance; the proposed method has less generalizability but with much lower variance. We have added the following discussion into Related Work.
> “…as the Rep gradient (Grathwohl et al., 2017). SCG (Schulman et al., 2015) utilizes the generalizability of REINFORCE to construct widely-applicable stochastic computation graphs. However, REINFORCE is known to have high variance, especially for high-dimensional problems, where the proposed methods are preferable when applicable (Schulman et al., 2015). Stochastic back-propagation…”
>
> Thank you for pointing out these fundamental conditions, which we have added into the revised manuscript.
>
> We hope your concerns have been addressed. If not, further discussion would be welcomed.

---

### Official Review · AnonReviewer2 · 2018-11-02
**Ambitious paper addressing a relevant problem. But not clear the novel contributions. High overlap with previous papers.**

**Rating:** 7
**Confidence:** 4

**Review:**

This paper presents a gradient estimator for expectation-based objectives, which is called Go-gradient. This estimator is unbiased, has low variance and, in contrast to other previous approaches, applies to either continuous and discrete random variables. They also extend this estimator to problems where the gradient should be "backpropagated" through a nested combination of random variables and a (non-linear) functions. Authors present an extensive experimental evaluation of the estimator on different challenging machine learning problems.


The paper addresses a relevant problem which appears in many machine learning settings, as it is the problem of estimating the gradient of an expectation-based objective. In general, the paper is well written and easy to follow. And the experimental evaluation is extensive and compares with relevant state-of-the-art methods.

The main problem with this paper is that it is difficult to identify its main and novel contributions.

1. In the case of continuous random variables, Go-gradient is equal to Implicit Rep gradients (Figurnov et al. 2018) and pathwise gradients (Jankowiack & Obermeyer,2018). Furthermore, for the Gaussian case, Implicit Rep gradients (and Go-gradient too) are equal to the standard reparametrization trick estimator (Kingma & Welling, 2014). This should be made crystal-clear in the paper. What happens is that the authors arrive at this solution using a different approach.

In this sense, claims about the low-variance of GO-gradient wrt to other reparametrization baed estimators should be removed, as they are the same. Moreover, I don't think some of the presented experiments are necessary. Simply because for continuous variables similar experiments have been reported before (Figurnov et al. 2018, Jankowiack & Obermeyer,2018).

2. It seems that the main novel contribution of the paper is to extend the ideas of (Figurnov et al. 2018, Jankowiack & Obermeyer,2018) to discrete variables. And this is a relevant contribution.  And the experimental evaluations of this part are convincing and compare favourably with other state-of-the-art methods.

3. Authors should be much more clear about which is their original contribution to the problems stated in Section 4 and Section 5. As authors acknowledge in Section 6. <<Stochastic back-propagation (Rezende et al., 2014; Fan et al., 2015), focusing mainly on re-parameterizable Gaussian random variables and deep latent Gaussian models, exploits the product rule for an integral to derive gradient backpropagation through several continuous random variables.>> This is exactly what authors do in these sections. Again it seems that the real contribution of this paper here is to extend this stochastic back-propagation (Rezende et al., 2014; Fan et al., 2015) ideas to discrete variables. Although this extension seems to be easily derived using the contributions made at point 2.

Summarizing, the paper addresses a relevant problem but they do not state which their main contributions are, and reintroduce some ideas previously published in the literature.

---

> ### Author Response · Authors · 2018-11-14
> **Addressing Reviewer 2 concerns**
>
> Thank you for your time and effort of reviewing our paper. Please see our response below.
>
> Our main contributions include:
>
> (i) For single-layer random variables (RVs), we propose a unified gradient named GO by exploiting the integration-by-parts idea, which is applicable to continuous/discrete RVs. In the special case of single-layer continuous RVs where GO recovers Implicit Rep or pathwise gradients, we consider it’s our contribution to provide a principled explanation (via integration-by-parts) why Implicit Rep and pathwise gradients have low Monte Carlo variance; or in other words, we prove that their implicit differentiation originates from integration-by-parts.
>
> (ii) For multi-layer RVs, our main contribution is the discovery that with GO (or in other words, the introduced variable-nabla), one can back-propagate gradient information through a nested combination of nonlinear functions and general RVs (including non-reparameterizable continuous RVs, back-propagating through which is challenging). Another interpretation of this contribution is that GO enables generalizing the deterministic chain rule to a statistical version. Here, we refer to deterministic chain rule as back-propagating gradient through deterministic functions (like neural networks) or reparameterizable RVs (like Gaussian). By contrast, statistical chain rule is referred to as back-propagating gradient through more general RVs (including non-reparameterizable ones). Of course, statistical chain rule recovers deterministic chain rule for deterministic functions and reparameterizable RVs, because GO recovers the standard Rep.
>
> (iii) Another 2 minor contributions include Lemma 1 and Corollary 1. In Lemma 1, we explicitly prove that our deep GO gradient contains the standard Rep as a special case, in general beyond Gaussian. Note neither Implicit Rep nor pathwise gradients can recover Rep in general, because a neural-network-parameterized reparameterization usually leads to a nontrivial CDF. In Corollary 1, we reveal the fact that the proposed method degrades into the classical back-propagation algorithm under specific settings.
>
> Finally, we believe it is interesting to create a consistent architecture, which unifies (a) a GO gradient which contains many popular gradients as special cases, and (b) a more general statistical chain rule developed based on GO which recovers the well-known deterministic chain rule under specific cases.
>
> For your comments not addressed above, please see our additional response below.
>
> (1) We have made clearer the relationships among the standard Rep, Implicit Rep/pathwise, and our GO in the revised manuscript. In the revised paper we have explicitly pointed out that the experiments from (Figurnov et al. 2018, Jankowiack & Obermeyer,2018) additionally support our GO in the special case of single-layer continuous RVs.
>
> (2) Please refer to our main contributions summarized above, where other contributions, beyond GO for discrete RVs, are clarified.
>
> (3) Please refer to our main contributions (ii)-(iii). As stated in our paper, many works tried to solve the problem of stochastic/statistical back-propagation. We consider our contributions in Secs. 4 and 5 as one step toward that final goal. Please note that what’s done in Secs. 4 and 5 is not straight-forward and has not been reported before. Since stochastic back-propagation (Rezende et al., 2014; Fan et al., 2015) focuses mainly on reparameterizable RVs, deterministic chain rule as mentioned in main contribution (ii) can be readily applied. By contrast, we target towards more general situations in Secs. 4 and 5 where deterministic chain rule might not be applicable, such as for non-parameterizable (continuous) RVs. We prove that one can utilize our GO to sequentially back-propagate gradient though non-parameterizable continuous RVs, namely the statistical chain rule mentioned in main contribution (ii).
>
> We have revised the last paragraph of the Introduction to make a more explicit summation of our main contributions, as mentioned above.
>
> We hope your concerns have been addressed. If not, further discussion would be welcomed.

---

### Public Comment · (anonymous) · 2018-10-01
**Questions**

1) Implicit reparameterization gradients (Jankowiak & Obermeyer 2018, Figurnov et al. 2018) already show improvements over GRep and RSVI, so it would seem natural to use them as the baseline for Sec 7.1. In this setting, what is the relationship between GO and Implicit Reparameterization Gradients.

2) In Sec 7.2, GO gradients require evaluating f many times. It would seem natural to compare to Local Expectation gradients in this case. What is the relationship between GO and LEgrad in this case?

3) For the discrete case, because we are making many calls to f, would it make sense to compare to multisample techniques (e.g., VIMCO)?

4) ARM (Yin 2018) is a recent technique for discrete random variables that uses multiple function evals. What is the relation with GO gradients?

---

> ### Author Response · Authors · 2018-10-01
> **Response to Questions**
>
> Thanks for your interest. Your comments are addressed below.
>
> (1) Similar to GO, Implicit Reparameterization (ImplicitRep) Gradients (Jankowiak & Obermeyer 2018, Figurnov et al. 2018) tried to exploit the gradient information of function f(z) for lower Monte Carlo variance, via a technique they termed implicit differentiation. Although seeming different, ImplicitRep is more or less a special case of GO in the single-layer continuous situation (thus no need for comparison). One can reveal this by comparing their Eq. (5) with our Eq. (9) in Theorem 1. The difference is that GO generalizes to discrete situations (Theorem 1), and also to deep probabilistic graphical models (Theorem 2 and 3).
>
> (2) As stated in the paragraph before Section 4, we adopt the local expectation idea when it is applicable and computationally acceptable. In some specific cases, like discrete random variables with finite support, fully applying the local expectation idea will reduce GO to the LEgrad. However, GO has the advantages that it is applicable to discrete situations with (1) infinite support (where LEgrad may not be applicable); (2) finite support (where LEgrad may be computationally expensive).
>
> (3) Thank you for your suggestions. We plan to fully exploit (and potentially improve) GO under various (discrete) cases in the future. However, we consider it beyond the scope of this conference paper, which is meant for presenting the derivation of a unified gradient that is widely applicable.
>
> (4) ARM (Yin 2018), using techniques (including data augmentation, permutation, and variance reduction) to aid REINFORCE for gradient calculation, is applicable to discrete situations with finite support. By comparison, GO, motivated by the connection of REINFORCE and Rep, is (1) a widely applicable gradient (continuous or discrete); (2) can be applied to discrete situations with infinite support. There might be some implicit relations between ARM and GO. We leave that as future work.

---

### Author Response · Authors · 2018-11-14
**Revision uploaded**

We thank all the reviewers for their time and effort.

We have responded to each of the reviewers and have uploaded a revised manuscript which addresses the reviews and comments.

Further discussion would be welcomed.

---

### Public Comment · ~Wu_Lin2 · 2019-06-18
**Weakening the smoothness condition of the implicit reparameterization gradients**

You may look at our poser  https://github.com/yorkerlin/VB-MixEF/blob/master/poster_workshop.pdf for the ICML workshop on Stein's method, where we weaken the smoothness assumption of the implicit reparameterization gradients. In other words, we also weaken the smoothness condition of the GO gradient for continuous cases.  In our poster, we only focus on the exponential family. However, the idea can be readily extended to general continuous univariate distribution. For multivariate case, you can use Theorem 4 in our poster.

---

> ### Author Response · Authors · 2019-06-19
> **Thanks a lot for the reminder**
>
> Thanks for reminding your interesting work. We agree that those smoothness assumptions mentioned would help with a more rigorous mathematical foundation for the GO gradient. Also, it would be appreciated if you could mention our paper like in your Theorem 3.
>
> A quick question about weakening the smoothness condition of the GO gradient: in your Theorem 4, what's the definition of $\nabla_{z_j} h(z)$ when $h(z)$ is not continuously differentiable? Thanks.

---

> > ### Public Comment · ~Wu_Lin2 · 2019-07-11
> > **Definition of $\nabla_{z_j} h(z)$**
> >
> > Please see Definition 1.  For example, if h(z) is a Relu function, the current implementation for computing the gradient is correct since the gradient exists almost everywhere except the null set {0}.

---

### Meta-Review · Area_Chair1 · 2018-12-18
**A comprehensive mathematical framework for unbiased low variance gradient estimator that applies to continuous and discrete random variables**

**Confidence:** 4
**Recommendation:** Accept (Poster)

**Metareview:**

This clearly written paper develops a novel, sound and comprehensive mathematical framework for computing low variance gradients of expectation-based objectives. The approach generalizes and encompasses several previous approaches for continuous random variables (reparametrization trick, Implicit Rep, pathwise gradients), and conveys novel insights.
Importantly, and originally, it extends to discrete random variables, and to chains of continuous random variables with optionally discrete terminal variables. These contributions are well exposed, and supported by convincing experiments.
Questions from reviewers were well addressed in the rebuttal and helped significantly clarify and improve the paper, in particular for delineating the novel contribution against prior related work.